# Integrative transcriptomic analysis reveals oligodendrocyte lineage switching in multiple sclerosis

Zhong-Ze Yan[1,2,3,4,*], Pei-Pei Liu[1,3,4,*], Hong-Zhen Du[1,3,4], Guo-Liang Chai[5], Zhao-Qian Teng[1,2,3,4], Chang-Mei Liu[1,2,3,4]

**Multiple sclerosis (MS) is a chronic disease of the central nervous system. The occurrence of MS is a phased process while its cause is still unclear. Here, by combining white matter single-nucleus transcriptomic datasets from MS and control samples, we found molecular crosstalk between oligodendrocytes (OLs) and immune cells involved in MS pathology. Using a machine learning approach, we identified oligodendrocyte precursor cells (OPCs) and OL subtypes at various developmental stages. We highlighted their unique molecular characteristics and analyzed their distribution throughout development, adulthood, and in different regions impacted by MS. We also found an increased number of Pre-OPCs and newly formed oligodendrocytes (NFOLs) in normal appearing white matter (NAWM), which were scarcely detected in MS lesions. By cell communication analysis and in vitro coculture, we found the interaction between SIRPA on microglia and CD47 on stressed oligodendrocytes was significantly reduced in MS lesions compared with NAWM, potentially preventing microglial phagocytosis of OLs. In contrast, CD74-MIF signaling between microglia and OLs was increased in lesions, which may lead to their retention around OLs.**

## Introduction

Multiple sclerosis (MS) is a chronic autoimmune disease affecting the central nervous system (CNS), marked by demyelination, axonal injury, and progressive neuronal loss (Lassmann, 2018). According to the World Health Organization (WHO), over 2.8 million individuals globally are living with MS, and its incidence is steadily rising (Walton et al, 2020). In MS, demyelinated axons in the CNS reduce or block the conduction velocity of electrical impulses to and from the brain, due to the formation of plaques in multiple regions (McDonald & Sears, 1969). However, the underlying mechanisms of inflammation and demyelination across different brain regions in MS remain unclear. Moreover, individuals with MS often exhibit a diminished capacity for remyelination-the regeneration of myelin-and the reasons for this impaired regenerative process are still not fully elucidated.

Oligodendrocytes (OLs), the myelin-forming cells of the CNS, are derived from oligodendrocyte progenitor cells (OPCs). Mature OLs extend their specialized membranes to wrap around axons, forming compact myelin sheaths that facilitate rapid nerve signal transmission. Importantly, OPCs, newly generated OLs and mature OLs, display substantial heterogeneity and exhibit region-specific alterations in cellular composition and lineage clusters across different lesions in MS. A study by Jäkel et al (2019) revealed distinct OL subclusters in MS, identifying intermediate state between OPCs and mature OLs, as well as terminally differentiated myelinating clusters. Both MS lesions and normal-appearing white matter (NAWM) showed reduced numbers of OPCs and intermediate OLs, whereas mature OLs in NAWM up-regulated myelin-related genes, suggesting enhanced remyelination activity. This altered OL heterogeneity in MS may be crucial for understanding disease progression and developing effective therapeutic strategies.

As MS progresses, immune cells are recruited to sites of ongoing lesions, where they contribute to myelin damage and further exacerbate disease pathology. It is believed that disease develops when inflammatory and other immune cells become dysregulated, shifting from physiological surveillance to a pathological immune response (Compston et al, 2008; Dolei et al, 2014). Although our understanding of how peripheral immunity contributes to inflammation in MS has advanced over the past decade, the interactions between immune cells and other CNS-resident cell types during neuroinflammation remain poorly understood. Microglia, the resident immune cells of the brain (Mariani & Kielian, 2009), play key roles in host defenses against pathogens and in maintaining CNS homeostasis by clearing debris such as dead cells, damaged neurons, and protein aggregates (Michell-Robinson et al, 2015). However, dysregulated microglia activation can lead to excessive neuroinflammation, which is a critical driver of neurodegenerative disease progression. Therefore, elucidating the

---

[1]Key Laboratory of Organ Regeneration and Reconstruction, State Key Laboratory of Stem Cell and Reproductive Biology, Institute of Zoology, Chinese Academy of Sciences, Beijing, China    [2]Savaid Medical School, University of Chinese Academy of Sciences, Beijing, China    [3]Institute for Stem Cell and Regeneration, Chinese Academy of Sciences, Beijing, China    [4]Beijing Institute for Stem Cell and Regenerative Medicine, Beijing, China    [5]Department of Neurology, Xuanwu Hospital, National Center for Neurological Disorders, Capital Medical University, Beijing, China

Correspondence: tengzq@ioz.ac.cn; liuchm@ioz.ac.cn
*Zhong-Ze Yan and Pei-Pei Liu contributed equally to this work

differences between OL and immune cell subsets across various lesion regions—and the regulatory mechanisms underlying their interactions—is essential for understanding MS progression.

In this study, we performed an integrative single-nucleus RNA sequencing (snRNA-seq) analysis of brain samples from 46 MS patients and 30 healthy controls to investigate cell-type heterogeneity and immune responses across distinct lesion regions. Our findings revealed increased abundance of stressed OLs, immune-associated OLs (ImOLs) and microglia in MS lesions. Using a machine learning-based approach, we identified distinct subtypes of OPCs and OLs corresponding to different stages of differentiation and maturation. Pre-OPCs and NFOLs were abundantly detected in NAWM but were nearly absent in the cores of CA and CI lesions, and only sparsely present at lesion edges—where their abundance was lower than in healthy controls. Mature oligodendrocytes (MOLs) were found in both NAWM and lesion edges but were significantly reduced in lesion cores. These results suggest that during MS progression, the regenerative capacity of OPCs and OLs is insufficient to replace the OLs lost through demyelination. In MS samples, differentially expressed genes (DEGs; $P$-values < 0.05) in Pre-OPCs were significantly enriched in glial development and differentiation function. DEGs in NFOLs were largely enriched in oxidative stress response, indicating heightened immune activation and inflammation. Furthermore, we confirmed that microglia (MG) were strongly activated in MS lesions, particularly at lesion edges, and interacted closely with stressed OLs and ImOLs. Finally, we identified SIRPA-CD47 and CD74-MIF as key molecular mediators of intercellular communication between MG and OLs, promoting OL loss and contributing to CNS pathology in MS. In summary, our study uncovers molecular mechanisms underlying demyelination across different lesion regions, offering potential targets for therapeutic intervention in MS.

## Results

### Integrative single-nucleus RNA sequencing reveals an increase of stressed OLs at the lesion edges, accompanied by a decrease of OPCs

To systematically investigate the pathological mechanisms underlying MS, we analyzed previously published snRNA-seq datasets for MS brain samples with well-documented clinical metadata, including lesion region information and age-matched healthy controls. All datasets were retrieved from the Gene Expression Omnibus (GEO) database. As described above, MS samples were classified into multiple lesion-associated regions- chronic active lesion (CA), CA edge, chronic inactive lesion (CI), CI edge, and NAWM-based on established criteria from a previously published study (Lassmann et al, 1998) (Fig 1A). Metadata with information on lesion regions, clinical ages, genders, and disease stages were also obtained (Table S1). After quality control, a total of 161,643 cells and 48,057 distinct genes were retained for downstream single-cell transcriptomic analysis. An anchor-based integration method was applied to merge 76 samples from five independent datasets (Fig

S1A). Principal component analysis (PCA) was performed for dimensionality reduction, followed by clustering using the Louvain algorithm, which identified 26 initial cell clusters. Cell type annotation was performed using classical marker genes, and further validated using a well-annotated snRNA-seq dataset from the healthy human brain (Lake et al, 2018). Detailed cluster annotations, including the top 100 DEGs for each cluster, are provided in Table S2. In total, nine major cell type clusters were identified: excitatory neuron (Ex-Neuron), inhibitory neuron (In-Neuron), OL, OPC, immune cell (Imm), MG and macrophages (MP), astrocyte (AST), endothelial and pericyte cell (Endo & Per), and vascular cell (Vasc) (Figs 1B and C and S1B).

To investigate cell-type alterations during MS progression, we applied the Milo algorithm (Dann et al, 2022) (see the Materials and Methods section) to quantify differential abundance of cell populations between healthy white matter and MS lesion regions. The AST2 and Vasc clusters were excluded from downstream analysis due to having fewer than three replicates. We then assessed the abundance of nine major cell types. Compared with healthy white matter, MS lesions exhibited marked differences in cell population composition (Fig 1D). Notably, the proportions of MG, AST, and multiple immune cell subsets were elevated across most MS lesion regions relative to controls (Fig 1E), consistent with previous study (Absinta et al, 2021). Using an inflammatory gene set (Table S3A), we observed a significant increase in inflammatory scores within MG/MP and Imms in NAWM, CA edge, and CI edge regions (Fig S1C and D). These findings suggest that immune activation is already evident at lesion edge of MS brains. Moreover, the reduced presence of OPCs in these regions may contribute to insufficient remyelination.

To construct a detailed map of OL subtypes, we categorized OLs into four groups: OPCs, normal OLs, stressed OLs, and ImOLs (Fig 1F). As previously reported (Jäkel et al, 2019; Absinta et al, 2021), stressed OLs are characterized by elevated expression of genes related to unfolded protein binding and heat-shock proteins, reflecting cellular stress induced by inflammation and injury. ImOLs were defined by co-expression of canonical OL marker genes along with immune-related genes such as APOE, CD74, and ITPR2. The full list of marker genes used to define each OL subtype is provided in Table S3B. Our analysis revealed that the abundance of both stressed OLs and ImOLs increased in most MS lesion regions (Fig 1G). To examine the effect of imbalanced OL subtypes on MS-related demyelination, we performed differential gene expression analysis between normal and stressed OLs across different MS regions (Table S4). In NAWM, stressed OLs displayed more than 1.5-fold up-regulation of DEGs compared with normal OLs in control, with significant enrichment in the KEGG pathways "Efferocytosis" and "Necroptosis" ($P$ values < 0.05; Table S5). Gene ontology (GO) terms associated with these DEGs included "immune response-regulating cell surface receptor signaling pathway involved in phagocytosis," "Fc-gamma receptor signaling pathway involved in phagocytosis" and "integrin-mediated signaling pathway" (adjusted $P$-values < 0.05; Table S6). Furthermore DEG analysis comparing stressed OLs in NAWM versus controls (adjusted $P$-values < 0.05, Table S4G) showed enrichment of GO and KEGG terms such as "cell junction assembly," "integrin-mediated signaling pathway" (Table S6M), and "Adherens junction" (Table

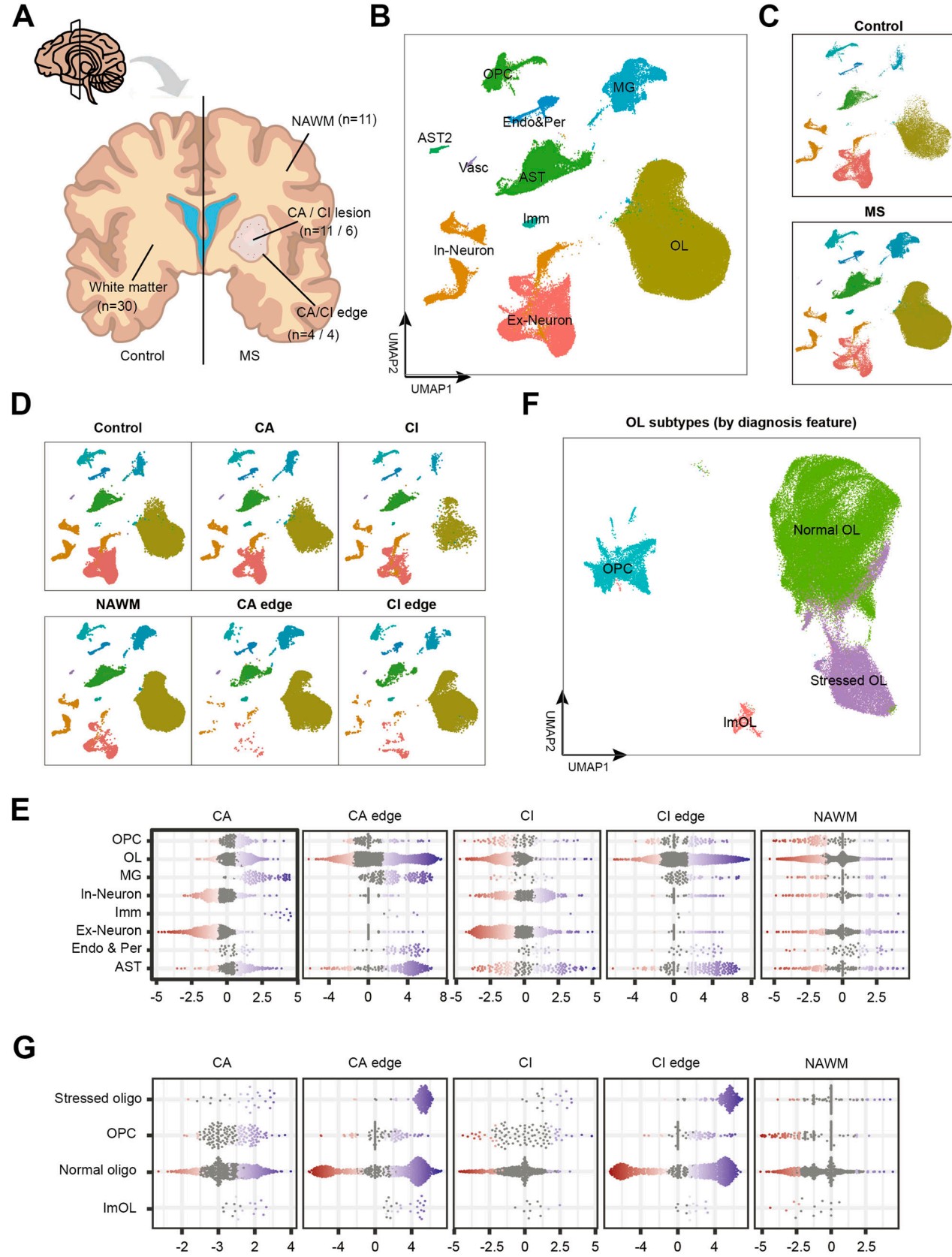

S5M), suggesting that stressed OLs in NAWM exhibit enhanced stress responses to the inflammatory microenvironment compared with healthy white matter. In addition, stressed OLs in CA, CI, and their edge regions showed up-regulation of genes related to the "Response to unfolded protein" and "regulation of autophagy and apoptotic," indicating region-specific stress responses. DEGs analysis of immune OLs in MS versus healthy controls (Table S7) revealed that ImOLs in NAWM were enriched for genes involved in inflammatory mediator regulation, including ASIC and ITPR1. These findings suggest that abnormal OL subtypes in MS exhibit impaired maturation, enhanced inflammatory signaling, and dysregulated apoptotic responses, all of which may contribute to disease progression.

## Oligodendrocyte heterogeneity identification by machine learning

For a long time, OLs were considered a functionally homogeneous population within the CNS. However, with the development of single-cell sequencing technologies, increasing evidence has confirmed the existence of oligodendrocyte heterogeneity (Marques et al, 2016; Wang et al, 2023). As the brain matures, genes specifically expressed during early stages of OPC and OL differentiation become less detectable. Consequently, in the adult human brain and in MS patients, it becomes increasingly difficult to distinguish OPC and OL subtypes at different differentiation stages using unsupervised clustering methods alone. To accurately characterize OL differentiation and maturation trajectories in adult humans, we applied a prototype-based supervised machine learning approach (Figs 2A and S2A–D) (La Manno et al, 2016). Several previously published human brain developmental atlases were used as training datasets (Herring et al, 2022; Su et al, 2022; van Bruggen et al, 2022; Braun et al, 2023) (Table S8, Fig S2E). These datasets spanned from fetal development to adolescence, a period during which OPCs and OLs are actively generated and undergo differentiation. To ensure consistency with known developmental OL subtypes, we selected prototype cells based on high-confidence, experimentally validated marker genes (Rivers et al, 2008; Marques et al, 2016; Fan et al, 2020). Specifically, the following marker genes were used to define each OL subtype: pre-OPC (*EGFR, NES, ZFP36L1,* and *GFAP*), dividing-OPC (*MKI67, ASPM, TOP2A,* and *CDK1*), late-OPC (*SOX10, NKX2-2, PCDH15,* and *CSPG4*), committed oligodendrocyte progenitors (COPs [*NEU4, SOX6,* and *GPR17*]), newly formed oligodendrocytes (NFOLs [*TCF7L2, ITPR2, TMEM2,* and *GPR17*]), myelinating oligodendrocytes (MFOLs [*MAL, MOG, PLP1, OPALIN,* and *SERINC5*]), and MOLs (*KLK6, APOD, FAR1,* and *PMP22*). Using the trained model, we scored every oligodendrocyte in the query human brain datasets across various age groups. Cells with a similarity score ≥0.70 were conservatively considered subtype matches. All identified OL subtypes showed strong concordance

with prior annotations, supporting the model's high accuracy in classifying OL developmental states. During early fetal development, we identified pre-OPCs along with a small number of mature OPCs and OLs. As development progressed, NFOLs and MFOLs began to emerge. In childhood, there was a substantial expansion of in COPs, NFOLs, and MFOLs, indicating active oligodendrocyte differentiation. In adolescence, the proportion of pre-OPCs declined, while COPs, MFOLs, and mature OLs (MOLs) increased, reflecting the transition toward a more MOLs population and ongoing myelination. Notably, only a small number of dividing-OPC was detected across all developmental stages, possibly due to resolution limitations in our model (Fig S2E).

To determine whether oligodendrocyte turnover exists at different stages of differentiation in adult MS patients, we applied the trained machine learning model to identify OL subtypes in MS lesions, NAWM and healthy controls. Each oligodendrocyte in the adult and MS brain samples was assigned a similarity score reflecting its correspondence to a defined OL subtype (Fig 2B). Pre-OPCs and NFOLs were abundantly detected in NAWM but were nearly absent in the cores of MS lesions (CA and CI) and found only in limited numbers at lesion edges, where their abundance was lower than that in healthy adult white matter. MOLs were detected in both NAWM and lesion edges but were significantly reduced within lesion cores. These patterns suggest that during MS progression, the regenerative capacity of OPCs and OLs is insufficient to compensate for the OLs loss caused by demyelination (Figs 2C and D and S3). Using the previously established similarity score threshold, we extracted OL subtypes from all adult brain samples (Fig 3A). Whereas most identified OL subtypes exhibited preferential expression of their corresponding marker genes (Fig 3B), COP and NFOL populations predicted by the model showed weaker expression of early developmental markers. This highlights the current limitations in resolving intermediate states along the continuous OL differentiation trajectory. Taken together, these findings suggest that a failure to generate sufficient new OLs contributes to the demyelinating pathology observed in MS. Notably, pre-OPCs and NFOLs were enriched in NAWM but completely absent from MS lesion regions.

To identify potential mechanisms underlying defective myelination in MS, we performed DEGs analysis of OL subtypes-specifically pre-OPCs and NFOLs-in NAWM compared with control white matter (Fig 3C and D; Table S9). Among the 1.5-fold upregulated DEGs in NFOLs, we observed significant enrichment in the KEGG pathway "Endocytosis" and GO terms related to "negative regulation of protein localization," "negative regulation of nervous system development," and "negative regulation of gliogenesis" (adjusted *P*-values < 0.05; Fig 3E and F; Tables S10 and S12). These findings suggest disrupted OL differentiation and potential vulnerability to cell death in the MS brain (Lin & Stone, 2020). Similarly, the 1.5-fold upregulated DEGs in Pre-OPCs were enriched in pathways related to brain neurodevelopment and cognitive processes

---

**Figure 1. Location-specific oligodendrocyte heterogeneity between control and MS regions in white matter.**
**(A)** Pattern map of control white matter and MS regions (NAWM, CA, CA edge, CI and CI edge). All snRNA-seq data used in our study was collected from control white matter and these MS regions. **(B)** The UMAP projection of all cells (n = 161,643 nuclei from 30 control individuals and 29 MS patients), colored by their respective cell type. **(C)** snRNA-seq clustering split by disease condition. **(D)** snRNA-seq clustering split by control and MS regions, colored by cell type. **(E)** Beeswarm plots of differential brain cell types and oligodendrocyte subtypes abundance for MS regions versus controls. **(F)** Sub clustering and annotation for oligodendrocytes in UMAP plot.
**(G)** Beeswarm plots of differential brain cell types oligodendrocyte subtypes abundance for MS regions versus controls.

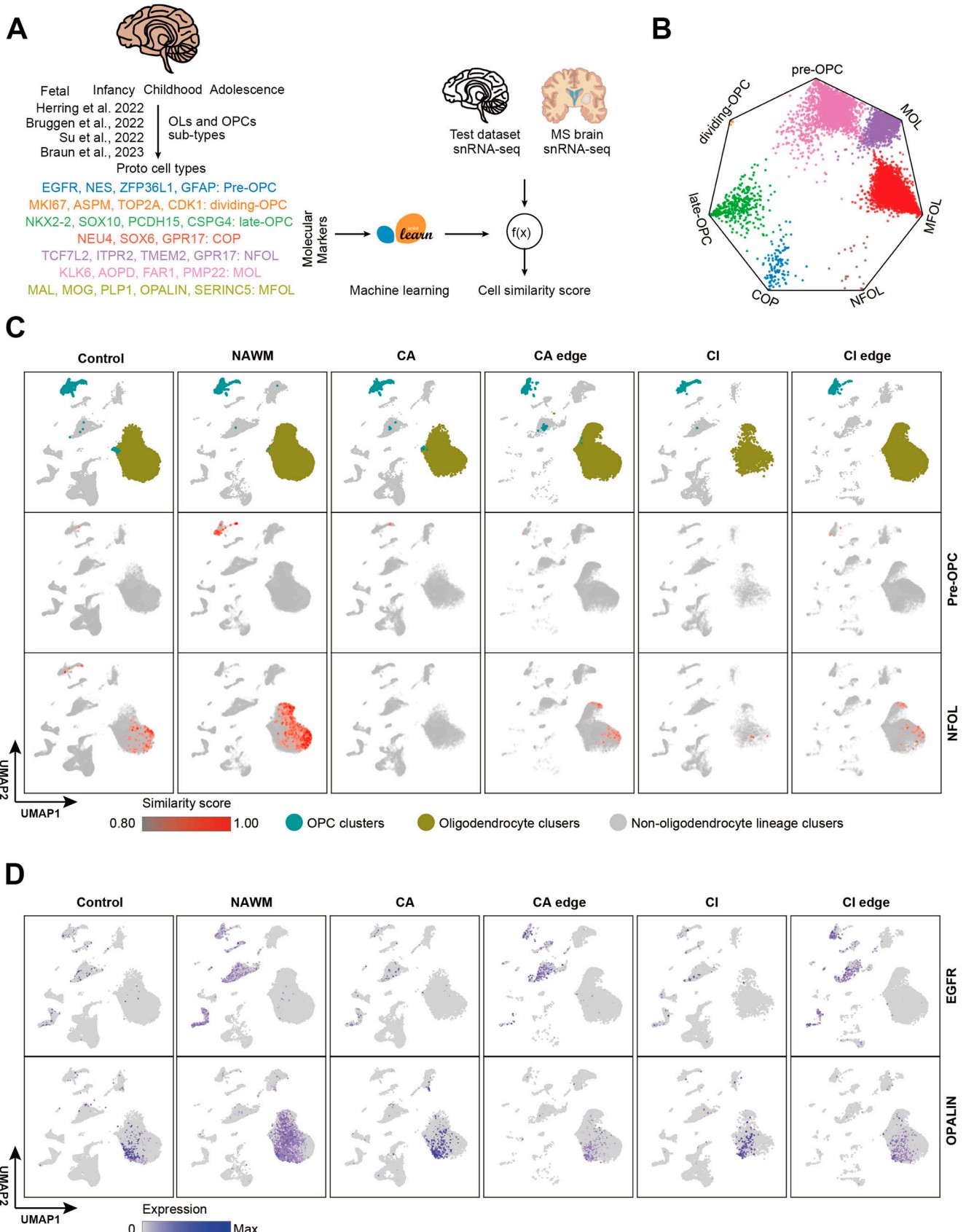

such as learning and memory (Adjusted *P*-values < 0.05; Fig 3E and F; Tables S10 and S12). Taken together, these results indicate that defective myelin formation in MS may be driven by an inadequate supply of newly generated oligodendrocytes, endoplasmic reticulum stress, and impaired processes related to membrane formation and cell adhesion.

### Inflamed microglia play a key role in immune-inflammatory activation in MS patients

To identify the role of immune cells in MS lesions, we calculated inflammatory scores for all cell types and immune-associated subtypes using an inflammatory gene set obtained from the MSigDB database (Liberzon et al, 2015) (Fig 4A and B; Table S3A). We found an obviously increased expression of inflammatory genes in MS regions, especially at MG and dendritic cells (DCs) in NAWM, CA, and CA edge (Figs 4C and S4A and B). Previous studies have shown that cells within chronic active lesions exhibit increased inflammatory activity (Luchetti et al, 2018). We hypothesized that this elevated inflammatory response might contribute to lesion expansion in the CA edge region. Using inflammation-associated marker genes (Liberzon et al, 2015) and inflammatory scores (Table S3B), we further classified MG into homeostatic MG (HMG) and inflamed MG (IMG) (Figs 4D and S4C and D). We found that IMGs, characterized by high inflammatory scores, were enriched in the NAWM, CA edge, and CA region, whereas HMGs predominated in the control and CI edge regions (Figs 4D and S4C and D). These findings support the idea that MG play an important role in MS lesion pathology, with IMGs in particular potentially driving lesion expansion at the CA edge.

### Interactions between OLs and MG are enhanced in lesion edge and NAWM

To better understand the pathophysiological processes in different regions of multiple sclerosis, we investigated the dynamics of intercellular communication among all CNS cell types. A ligand–receptor interaction-based approach was used to systematically construct cross-cell communication networks. We calculated the ligand–receptor interaction pairs between cell types across in six distinct lesion regions or disease conditions and found a significantly higher number of interactions in the none lesion region (NAWM, CA edge, and CI edge) than in the lesion region (CA and CI) (Figs 5A and S5A–F). In the non-lesion regions, astrocytes and IMG showed stronger interactions with OLs. Therefore, we further analyzed each ligand-receptor pair for OLs and immune-associated cells (AST, MG, DC, T cell, and B cell) (Table S11) and found significant pairs with high communication means. We found SIRPA-CD47 and CSF1R-CSF1 interactions are active in NAWM but lost in CA and CI lesion regions (Fig 5B). Previous studies

have shown that the SIRPA-CD47 interaction pairs can prevent cells from being phagocytosed by immune cells (Al-Sudani et al, 2023), and we speculate impaired SIRPA-CD47 interaction may contribute to MG mediated myelin phagocytosis in MS lesions. In addition, pairs associated with inflammatory cell migration, such as CD74-MIF, strongly interacted between IMG and OL lineages in CA and CI lesion regions, which kept a low level in control and NAWM samples (Fig 5B), suggesting the immune influence of MG on OL was significantly enhanced in these lesions.

### The enhanced interaction of SIRPA-CD47 signaling suppresses MG phagocytosis on OLs

CD47 signals through the SIRPA receptors (Morrissey et al, 2020). Our analysis of single-cell transcriptomic data revealed that CD47 expression was elevated on OLs in NAWM but markedly reduced in MS lesion regions, suggesting that this pathway may have important functions in disease progression (Fig 5C). To investigate the function of CD47 in regulating OL and MG responses in the context of MS, we overexpressed CD47 using a lentivirus-delivered vector carrying with GFP and assessed the impact of CD47-overexpression in MO3.13 OL lines (Fig S6A–C). In addition, we knocked down SIRPA using a lentivirus-delivered shRNA carrying with mCherry in MG HMC3 lines (Fig S6D–F). Compared with ctrl, all shRNAs resulted in a significant decrease in the expression of SIRPA protein in cultured microglia, and among them, shRNA2 had the highest efficiency (Fig S6D–F). To identify the role of CD47/SIRPA receptor-ligand interaction in vitro coculture experiment, we detected about 13% reduced phagocytosis by immunofluorescence assay after overexpression of CD47 in OLs coculture with MG cells and about 15% augmented phagocytosis after down-regulation of SIRPA in MG cells coculture with OLs (Fig 6A and B). To further confirm whether overexpression of CD47 or down-regulation of SIRPA could alter phagocytic activity, we assessed the rate of phagocytosis by FACS when cocultured with MG cells or OLs. Overexpression of CD47 in OLs repressed phagocytosis compared with control (Ctrl) and down-regulation of SIRPA in MG cells increased phagocytosis compared with Ctrl (Fig 6C–E). Together, these findings indicate that the SIRPA-CD47 axis mediates interactions between in OLs and MG and may enhance MG-mediated phagocytosis in MS lesions.

### The enhanced CD74-MIF interaction promotes MG accumulation

Our investigation of oligodendrocyte–microglia (OL–MG) interactions in MS also identified signaling mediated by macrophage migration inhibitory factor (MIF), a multifunctional protein involved in tissue repair and other biological processes (Su et al, 2017). In MS lesions, we found increased expression of the MIF receptor CD74 (Fig 5C). To explore the role of CD74-MIF mediated

**Figure 2. OL lineage identification in control and MS regions based on Prototype-Based Scoring.**
**(A)** Schematic illustration of the inference of oligodendrocyte subtype identity based on prototype scoring. **(B)** Wheel plot visualizing similarity scores of each cell in MS samples to predicted prototype by machine learning model. Dots represent every single cell whose distance to the angles of the polygon is proportional to the similarity score of that prototype. **(C)** UMAP plots showing snRNA-seq in control and MS regions colored by OL and OPC clusters (top row) and by similarity score to prototypical Pre-OPC (middle row) and NFOL (bottom row). **(D)** UMAP plots showing snRNA-seq in control and MS regions colored by Pre-OPC (EGFR, top row) and NFOL (OPALIN, bottom row) marker genes expression.

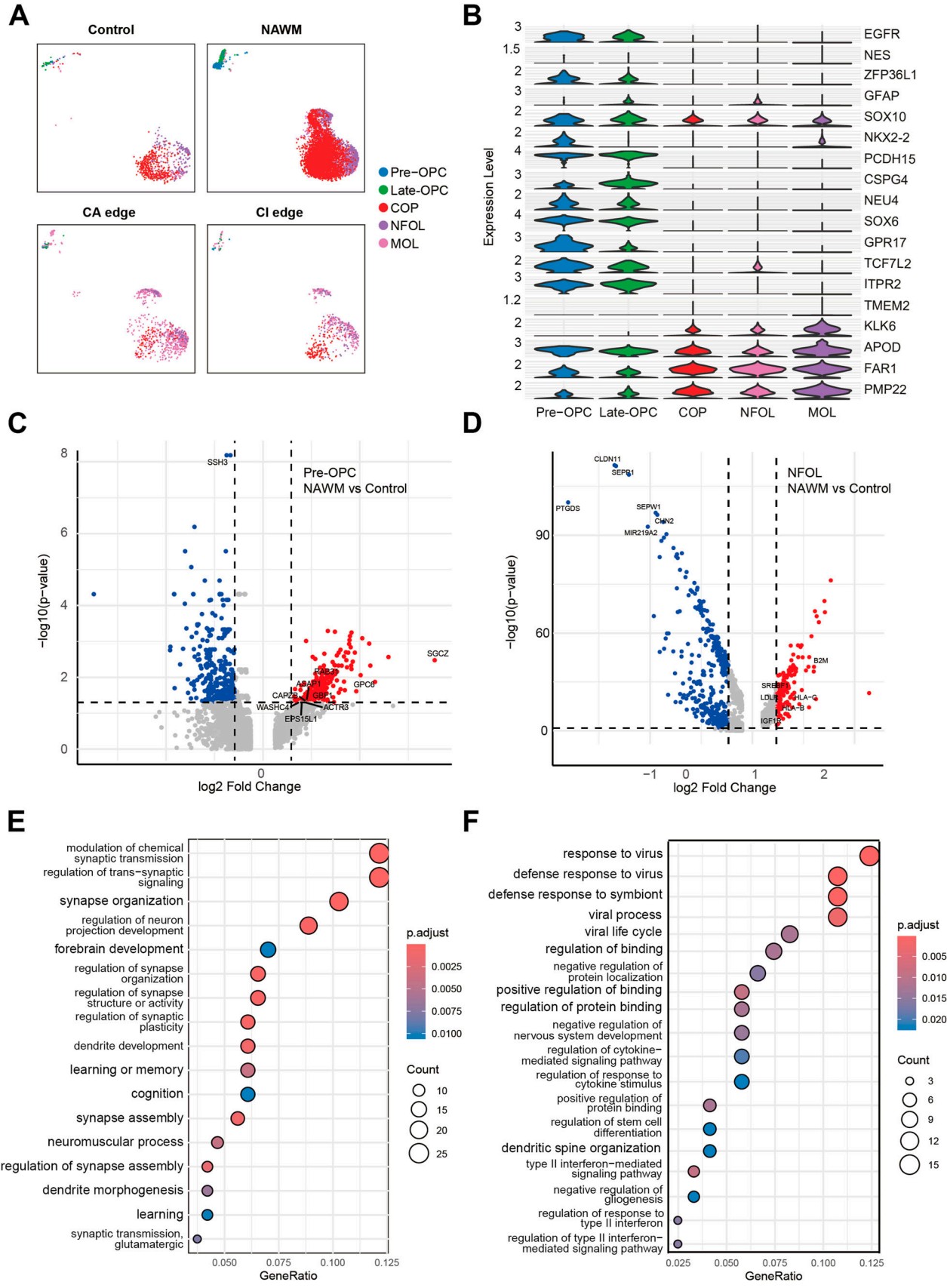

OL-MG interactions in tissue repair, we performed in vitro coculture experiments. MIF was overexpressed in OLs MO3.13 lines using a lentivirus-delivered vector carrying GFP (Fig S7A–C). CD74 was knocked down in MG HMC3 lines using a lentivirus-delivered shRNA construct carrying mCherry (Fig S7D–F). The efficacy of CD74 knockdown was confirmed by RT–PCR and immunofluorescence staining in MG HMC3 lines (Fig S7D–F). Using a transwell migration assay, we found that overexpression of MIF in OLs reduced MG migration (Fig 7A and B). However, MG with down-regulation of CD74 receptor showed increased MG migration activity (Fig 7A and B). Thus, signaling between OLs MIF and its receptor CD74 in MG may play a role in cell migration in MS.

# Discussion

In recent years, the widespread adoption of single-cell sequencing technologies has led to the generation of extensive datasets for the brain and various neurodegenerative diseases. MS, a neurological disease with wide-ranging implications, has been reported in several articles based on single-cell sequencing analyses revealing altered cellular components and cell differentiation fates. Selective cellular vulnerability is a hallmark feature of MS pathology (Falcão et al, 2018; Nataf et al, 2022). Previous studies have shown specific cell types that are especially vulnerable in MS, such as *CDH20*+ *RBFOX1*+ OLs and OPCs, compared with other neural cell types (Kirby et al, 2019; Nataf et al, 2022). However, the molecular signatures of selective OL vulnerability correlate with disease severity and the intensity of inflammation are largely unknown.

In this study, we showed how cellular heterogeneity manifests across different lesion regions in MS by integrating multiple single-nucleus RNA sequencing datasets. Using machine learning models, we identified subtypes of OL maturation and myelination without relying solely on traditional marker genes. We revealed that immature OPC precursors and OL populations associated with myelin formation were present in NAWM, but largely absent in lesion cores. Our findings suggested a stronger inflammatory response in the lesion edge and NAWM compared with the lesion core. These results suggest that OLs damage in MS may be alleviated by modulating the corresponding inflammatory response pathways, such as SIRPA-CD47 and CD74-MIF.

By integrating snRNA-seq with quantitative analysis of cell proportions across distinct MS lesion regions, we found that the number of normal OLs was reduced in each lesion region, whereas the proportion of stressed OLs significantly increased, especially in the lesion edges. OLs form myelin and provide nutritional and metabolic support to neurons by ensheathing axons with myelin

(Molina-Gonzalez et al, 2022). Stressed OLs activate the unfolded protein response, which can trigger inflammatory signaling pathways and lead to apoptotic cell death (Hetz et al, 2020; Absinta et al, 2021).

Accumulating evidence highlights the important role of OPCs in remyelination (Yun et al, 2022). Regional loss of OPCs has been demonstrated to result in demyelination, MG activation, and subsequent neuronal death in MS (Crawford Abbe et al, 2016; Nakano et al, 2017). Here, we found a decreased abundance of OPCs in all lesion regions except the CI region. Using a supervised machine learning approach, we accurately identified clusters corresponding to immature OPCs and newly forming OLs, characterized by the expression of genes associated with OL differentiation and maturation. These clusters were rarely detected in MS samples, particularly within lesion cores. Whereas most OL subtypes were robustly detected, the model showed limited resolution in distinguishing transitional populations such as COPs and NFOLs, which may affect the interpretation of their abundance in MS samples, particularly within lesion cores. Nevertheless, we observed a reduction in pre-OPCs, alterations in their gene expression profiles, and weakened cellular interactions in MS tissue. These findings suggest that early disruptions in OL lineage progression may impair remyelination in MS. Furthermore studies are needed to refine subtype resolution and investigate mechanisms underlying disease-related changes, particularly in human pre-OPC subtype.

MS is characterized by immune dysregulation that leads to the infiltration of immune cells into the CNS, triggering demyelination and neurodegeneration (Attfield et al, 2022; Rodríguez Murúa et al, 2022). Consistent with previous studies, our findings support the involvement of immune cells in interacting with OLs under both physiological and pathological conditions. Importantly, we found that the immune activity of MG and DCs tracks disease progression in MS. Previous studies showed that in MS patients, myelin destruction is associated with the activation of microglia, the resident innate immune cells of the CNS (Luo et al, 2017). In our analysis, we observed strong interactions between IMG and stressed OLs in NAWM. In addition, we found that stressed OLs are more vulnerable to MG-mediated attack, and that with disease progression, an increasing proportion of normal OLs transition into a stressed state, contributing to demyelination. Elucidating the intracellular mechanisms that drive the conversion from normal to stressed OLs will be critical for understanding disease progression and developing potential therapeutic strategies.

Our research found that increased presence of stressed OLs in MS lesions is associated with MS immune activation and demyelination, potentially impairing myelin maintenance (López-Muguruza & Matute, 2023). These OLs exhibited up-regulation of

**Figure 3. Altered heterogeneity of Pre-OPC and NFOL in control and MS regions.**
**(A)** UMAP plots showing the distribution of machine learning identified the OL lineage subtypes (Pre-OPC, Late-OPC, COP, NFOL and MOL) in control, NAWM, CA edge, and CI edge. **(B)** Violin plots showing the expression of feature genes for each of the OL lineage subtypes. **(C)** Volcano plot of DEGs of Pre-OPC between NAWM and control. Significant DEGs ($|\log_2$ fold change$| > 0.585$ and $-\log_{10}(P\text{-value}) > 1.5$) are highlighted in red dots. DEGs ($|\log_2$ fold change$| > 0.585$ and $-\log_{10}(P\text{-value}) > 1.5$) associated with OL development are marked as blue labels. **(D)** Volcano plot of DEGs ($|\log_2$ fold change$| > 0.585$ and $-\log_{10}(P\text{-value}) > 1.5$) of NFOL between NAWM and control. Significant DEGs ($|\log_2$ fold change$| > 0.585$ and $-\log_{10}(P\text{-value}) > 1.5$) are highlighted in red dots. **(E)** KEGG pathway enrichment analysis of DEGs of Pre-OPC ($P$-value $< 0.05$). **(F)** KEGG pathway enrichment analysis of DEGs of NFOL ($P$-value $< 0.05$).

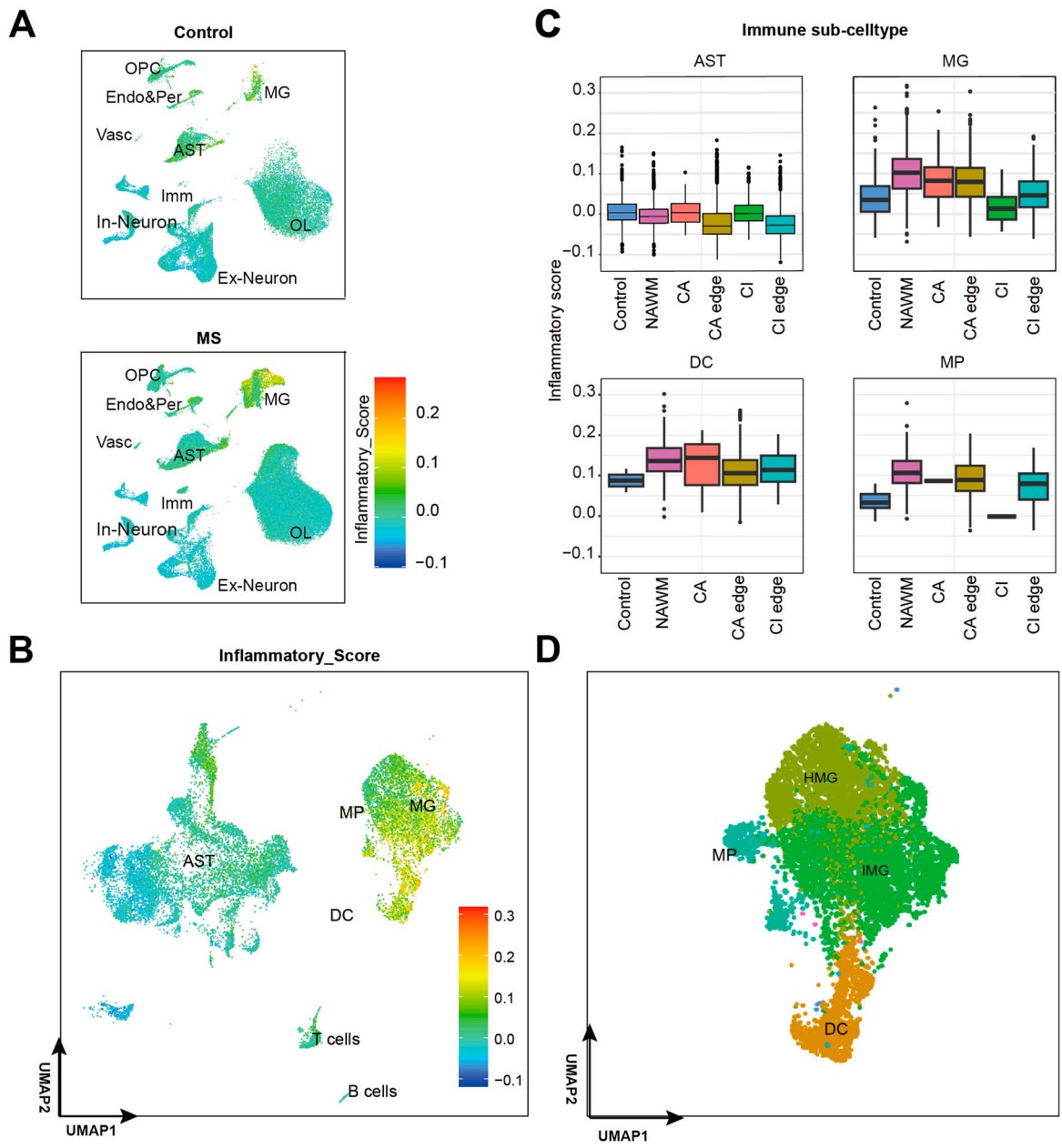

**Figure 4. IMG activated in MS condition.**
**(A)** UMAP plots of all cells colored by inflammatory score split by disease condition. **(B)** UMAP plot of immune associate cell types (AST, MG, DC, T cell and B cell) colored by inflammatory score. **(C)** Boxplots for the inflammatory score of AST, MG, DC, and MP in control and MS regions. **(D)** UMAP plot of MG sub-types (HMG and IMG), MP and DC.

autophagy and apoptosis-related pathways, suggesting impaired functionality in maintaining myelin integrity. However, the accumulation of stressed OLs may also result from environmental stressors induced by demyelination. Previous studies have shown immune-mediated attacks and complement activation can initiate, which subsequently leads to OL stress reaction (Watzlawik et al, 2010). Metabolic stress can also induce OL death, possibly as a response following demyelination (Fernandes et al, 2023). Taken together, this relationship appears to be bidirectional: stressed OLs may exacerbate demyelination, although at the same

time, they may arise as a consequence of inflammation and demyelination.

Using integrative analyses of snRNA-seq datasets combined with in vitro coculture assay, we also found that MG and OLs communication played a central role in MS pathology. To elucidate the cellular mechanisms active within lesions, we focused on the SIRPA-CD47 and CD74-MIF pairs, both of which mediate MG regulation of OLs during disease progression. SIRPA-CD47 pairs produce "do not eat me" signal protects synapses from excessive elimination during neurodevelopment (Ding et al, 2021; Jiang et al,

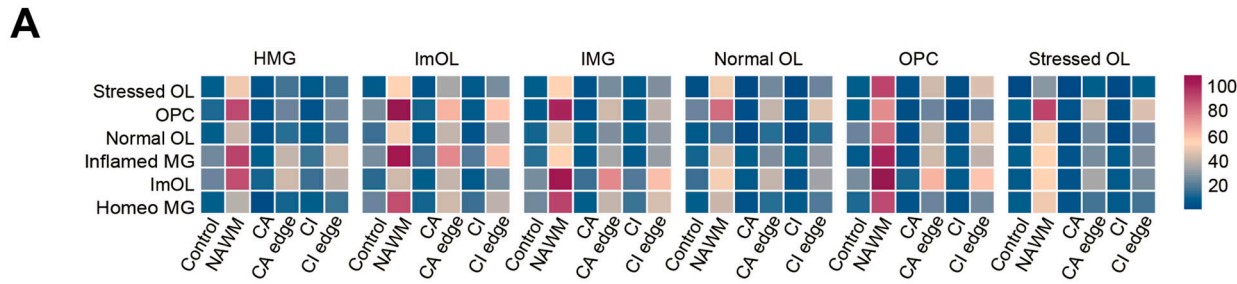

**Figure 5. The enhanced intercellular communications between OLs and IMG in MS NAWM and CA edge.**
**(A)** Heat maps show the cell communication level count between cell subtypes, in control and five MS regions. **(B)** Dot plot shows mainly interact pairs between OL and MG subtypes in control, NAWM and MS lesions. **(C)** Expression level plot for major ligand and receptor genes in control, NAWM and MS lesions.

2022). Our study indicates that reduced interactions between CD47 and SIRPA in both lesion cores and edges contribute to OL depletion and subsequent demyelination. Previous studies have also suggested that MIF, through its receptor CD74, can inhibit MG migration although promoting MG proliferation and inflammatory chemokine release, ultimately leading to forebrain cell death (Arnò

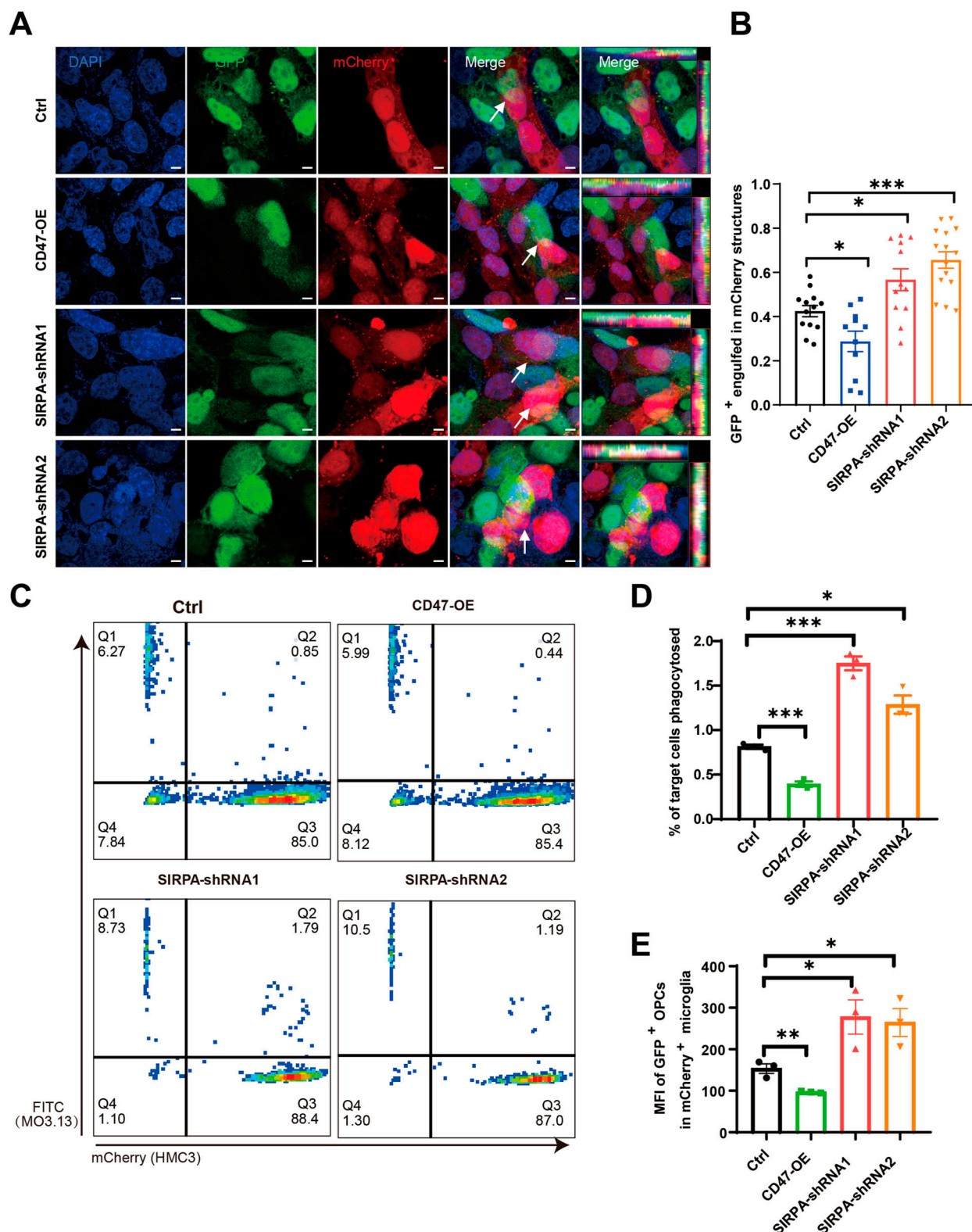

**Figure 6. SIRPA–CD47 receptor-ligand interaction to suppress MG phagocytosis.**
**(A)** representative image of OPCs (GFP+) cultured cocultured with MG (mCherry). Scale bar, 5 $\mu m$. **(B)** Analysis of GFP engulfed in mCherry + MGl structures (n = 11–15 OPCs per group). All data are mean ± SE. $P$-values determined by $t$ test. *$P$ < 0.05, ***$P$ < 0.001. **(C)** Flow cytometry phagocytosis plots showing rates of engulfment of OPCs (MO3.13) when cocultured with human-derived MG (labeled with HMC3). **(D)** Percentage of phagocytosed OPCs. All data are mean ± SE. $P$-values determined by $t$ test. *$P$ < 0.05, ***$P$ < 0.001. **(E)** MFI of GFP + OPCs in the mCherry + microglial population. All data are mean ± SE. $P$-values determined by $t$ test or one-way ANOVA. *$P$ < 0.05, **$P$ < 0.01.

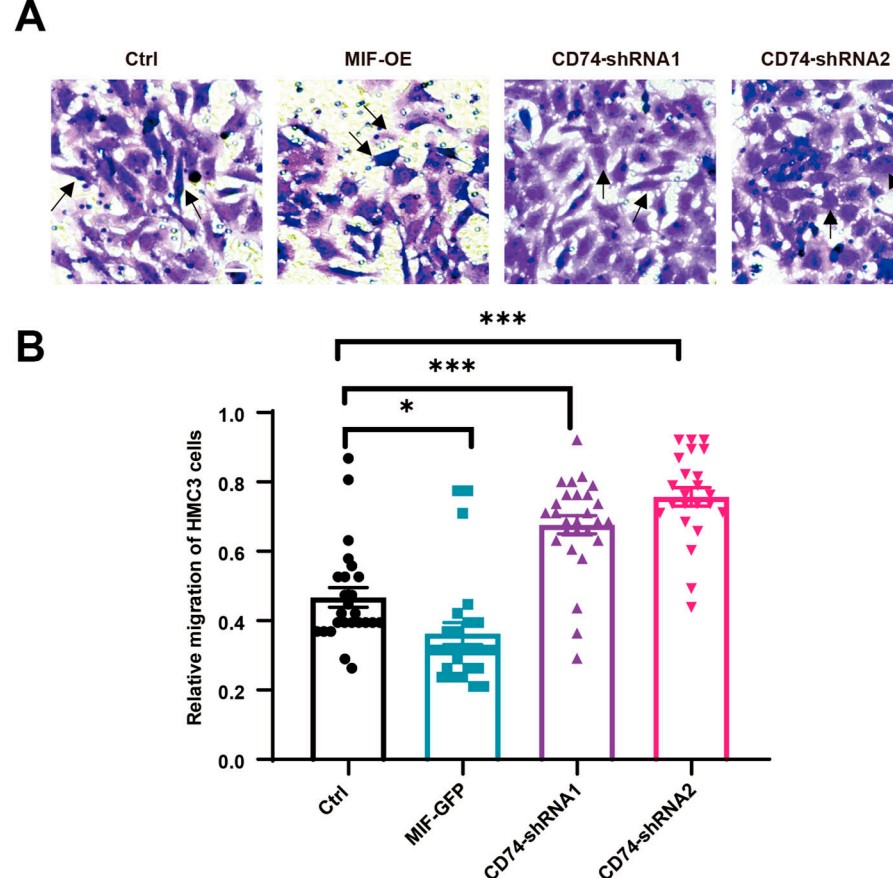

**Figure 7. CD74-MIF receptor-ligand interaction to inhibit MG migration.**
**(A)** Representative image of transwell cell migration assay. The migrating cells were stained with a violet color. The black arrow indicates cell staining. Scale bar, 50 μm. **(B)** Quantified the relative number of migrating cells of total MG cells. Migrated cell numbers in five fields (each field = 200 × 200 μm) were counted and the values were averaged. Three independent experiments were performed. All data are mean ± SE. *P*-values determined by *t* test. *P < 0.05, ***P < 0.001.

et al, 2014; Ghoochani et al, 2016; Jin et al, 2021). Here, we found MS lesion increased OPC-released MIF which bond to CD74 receptor and inhibited MG migration to evoke inflammation. Thus, we believe the CD74-MIF signaling axis inhibits the migration of MG and causes MG aggregation into the lesion core, resulting in loss of OLs, and ultimately demyelination. Our studies identify SIRPA-CD47 and CD74-MIF signaling as candidate targets for the therapeutic modulation of OLs and MG pathogenic activities in MS.

Targeting these pathways may offer a promising strategy for mitigating demyelination. Specifically, we propose that the down-regulation of CD47 in stressed OLs and OPC could be an adaptive mechanism to promote their elimination by MG, ultimately causing demyelination. The interaction involving CD74-MIF may initially inhibit microglial migration and promote the phagocytosis of stressed OLs. In future therapeutic applications, modulating the SIRPA-CD47 axis could help prevent excessive MG-mediated OL phagocytosis, thereby preserving myelin integrity. Similarly, fine-tuning the CD74–MIF pathway may allow for better control of inflammatory responses, enhancing cell survival, and promoting tissue repair.

A limitation of our study is that the interactions between OLs and MG were only validated through in vitro coculture assays. Future studies will provide a systematic analysis of using a new multiplex immunofluorescence and barcode tracing approach to validate SIRPA-CD47 and CD74-MIF signaling molecules in animal models and determine their potential as therapeutic targets for MS.

## Materials and Methods

### Data collection

Published human brain snRNA-seq data and corresponding sample information (Jäkel et al, 2019; Schirmer et al, 2019; Absinta et al, 2021; Kihara et al, 2022) were collocated for our bioinformatic analysis. A well-annotated human brain snRNA-seq data (Lake et al, 2018) was used as a reference and corresponding control. Metadata for all patient diagnosis information, including age, sex, clinical stage, and lesion region, were carefully organized.

### Data processing and integration

Seurat (v4.1.0) R package was used for snRNA-seq upstream analysis. Firstly, Matrix and metadata collected above were used to create Seurat objects, cells with <5% of mitochondrial and 200–6,000 genes expressed were retained. Count matrices were normalized using the log-normalization method with the NormalizeData function, applying a scaling factor of 10,000. The top

2,000 highly variable genes across cells were retained using the FindVariableFeatures function. An anchor-based approach, involving the FindIntegrationAnchors and IntegrationData functions, was used to combine all data into a single Seurat object although mitigating batch effects from different samples. After filtering and integration, 160,547 cells from 59 samples were included in our research.

## Cell clustering and cell type annotation

For these integrated data, each highly variable gene across cells was scaled by Seurat, and then principal component analysis (PCA) was used to reduce dimensions. The first 30 PCs were taken as input for edge weight between any two cells by the FindNeighbor function. Then all cells were grouped into 19 clusters by the FindClusters function with a resolution of 0.5. All clusters were visualized by UMAP.

Using classical marker genes (*PLP1*, *MBP*, *TMEM144*, *CNP*, *MOBP* and *MAG* for OL; PCDH15, *PDGFRA*, *BCAN*, *SOX6* and *OLIG1* for OPCs; *MAP2*, *GALNTL6*, *SYT1*, *ZMAT4*, *RELN* for neuron; *GFAP*, *AQP4*, *ADGRV1* for AST; *P2RY12*, *CX3CR1*, *CD74*, *C3* for MG and *CLDN5*, *ABCB1*, *ATP10A*, *FLT1* for Endo). 19 clusters were defined as major cell types, including OL, Neuron, MG, AST, and Endo. For neurons, we first resolved into ex-neuron, and in-neuron neuronal subtypes with classical marker genes *SLC17A7* and *GAD*. Then, using DEGs with > twofold change and *P*-values < 0.05, we identified seven ExNs and three InNs sub-cell types.

## Differential abundance analysis

Significant differences in cell type abundance between MS lesions and healthy controls were analyzed using the milo R pipeline (Dann et al, 2022). We tested differential abundance for mainly cell types and oligodendrocyte subtypes. We constructed a KNN graph using embedding k = 15.

## Different expression genes analysis and function enrichment

Difference expression genes between MS and control were detected by performing Wilcoxon tests by the FindMarkers function with adjusted *P*-values less than 0.05 and average $\log_2$ foldchange larger than 0.5. Using detected gene sets, gene ontology (GO) terms and Kyoto Encyclopedia of Genes and Genomes (KEGG) pathways were enriched using R package clusterProfiler (Yu et al, 2012). The annotation R package "org.Hs.eg.db" was used to trans-transform gene symbols and entrez gene IDs. The analysis output results were plotted by R package ggplot2.

## Cell scoring using machine learning method

To train a high-confidence model, we selected prototypes in the brain development period (childhood, adolescence, adult, and aging) and used this prototype as a reference to calculate similarity scores. Based on the L2-norm logistic regressed module, a multinomial machine learning method was used to train the log-transformed and normalized expression data. We used embryo and infant transcriptome datasets and used marker gene confirmed cell type cluster as prototype annotation as a train data. Prototypes include OPCs and OLs at different maturation and differentiated states (pre-OPC, dividing-OPC, later-OPC, COP, NFOL, MOL, and MFOL). To train the module, we used the top 4,500 highly variable genes as candidates. Then, we delimitated gene variation in all cell types and produced a more general model. The candidate gene number was further reduced.

## Immunoinflammatory and cytokine score analysis

To define inflammatory score, an inflammation-associated gene set named "HALLMARK_INFLAMMATORY_RESPONSE" was downloaded from MSigDB (Liberzon et al, 2015) (Tables S3 and S12). The addModuleScore function in the Seurat package was used to calculate inflammatory scores.

## Cell communication analysis

The Python project CellPhoneDB (Efremova et al, 2020) was used to predict cell-cell communication based on our single-cell analysis result. The normalized gene counts and the metadata containing cell types identified above were used as input files.

## Lentiviral constructs and production

CD47 or MIF overexpression lentiviral construct was made by integrating the PCR product of the human gene's open reading frame sequence into the pCD511B-copGFP vector (Youbio company). pCD511B-copGFP was used as the control. The primers were as follows:

CD47 forward, 5′- CTAGCTAGCATGTGGCCCCTGGTAGCGGCGCTGTT-3′; CD 47 reverse, 5′- CGCGGATCCTTATTCATTAAGGGGTTCCTCTACA-3′.

MIF forward, 5′- CTAGCTAGCATGCCGATGTTCATCGTAAACACCA-3′; MIF reverse, 5′- CGCGGATCCTTAGGCGAAGGTGGAGTTGTTCCAG-3′.

The sequence of shSIRPA or shCD74 was synthesized and subcloned into lentiviral vectors carrying with mCherry.

SIRPA-shRNA1: 5′-CTAGGCCGGGAATTAATCTACAATCTCAAGAGGATT GTAGATTAATTCCCGGCTTTTT-3′; SIRPA-shRNA2: 5′-CTAGGGAATGAG CTCTCAGACTTCCTCAAGAGGGAAGTCTGAGAGCTCATTCCTTTTT-3′. CD74 -shRNA1:5′-CTAGGCAGAATGCCACCAAGTATGGTCAAGAGCCATACTTGGT GGCATTCTGCTTTTT-3′; CD74-shRNA2: 5′-CTAGGCCACCAAGTATGGCAA CATGTCAAGAGCATGTTGCCATACTTGGTGGCTTTTT-3′.

For making lentivirus, we transfected lentiviral transfer vector DNA and packaged plasmid DNA into cultured 293T cells using polyethylenimine (PEI). The medium containing lentivirus was collected, filtered through a 0.22-$\mu$m filter, and concentrated.

## Construction of stably cell lines

To establish Ctrl, CD47, or MIF overexpression transfected cells, we used lentivirus into MO3.13 cell lines. After 48 or 72 h, the cells were harvested for FACs by green fluorescence positive cells. HMC3 cell lines infected with Ctrl, SIRPA-shRNA, CD74-shRNA lentivirus, and the cells were screened by FACs using mCherry.

## OPC-MG coculture

For OPC-MG cocultures, we first plated MG on glass coverslips coated with poly-D-lysine (P7405, 10 $\mu$g/ml; Sigma-Aldrich) at a density of 40,000–50,000 cells/well in 24-well plates and incubated overnight. After 24 h, we added OPCs in a 1:1 ratio into MG for 2 d. After 48 h, fixed cells were stained for DAPI and then quantified using a microscope.

## In vitro phagocytosis

The human MG cells (HMC3 line) were transfected with lentivirus carrying with mCherry. In a 3:1 ratio of MG (21 × 104 cells) to OPC cells (~7 × 104 cells), the cells were mixed in tubes and incubated at 37°C for 3 h on a shaker (14$g$) to promote MG and OPC cells interaction. Subsequently, the cells were washed and analyzed on the FACS machine.

## Flow cytometry analysis

OPCs (MO3.13 lines) infected viruses exhibit green GFP fluorescence. Microglia (HMC3 line) infected viruses exhibit red mCherry fluorescence. Flow cytometry analysis used the cell's fluorescence, no antibody staining, and no modulation compensation. The detailed flow cytometry analysis is as follows: The cells were harvested and washed twice with PBS. Immediately before flow cytometric analysis, the cells were strained through a 100-$\mu$M filter. Flow cytometry analyses were performed on the BD FACS Aria II (Becton Dickinson). Flow cytometry phagocytosis plots showing rates of engulfment of OPCs (labeled with GFP, FITC) when cocultured with HMC3 lines (labeled with mCherry, PE). Data were analyzed using FlowJo software (FlowJo).

## In vitro migration assay

The migration ability of the cells (HMC3) was assessed using 24-well Transwell (3422; Corning). The MO3.13 cells were placed into the lower chamber containing 10% FBS medium (1 ml), and 500 $\mu$l of serum-free medium with 5 × 104 HMC3 cells were added to the upper chamber. After 24 h, cells crossed the membrane were fixed with PFA and stained by crystal violet.

## Data Availability

The sequencing data sets supporting the conclusions of this article are available at the Gene Expression Omnibus (GEO; https://www.ncbi.nlm.nih.gov/geo/) repository, with accession numbers: GSE118257, GSE180759, GSE97942, GSE179590, and Sequence Read Archive (SRA; https://www.ncbi.nlm.nih.gov/sra/) repository, with accession numbers: PRJNA544731. The data used to train the machine learning model are download at GEO repository, with numbers: GSE168408, github.com/Castelo-Branco-lab/humandevOLG, https://github.com/linnarsson-lab/developing-human-brain. All analysis code used in this study is publicly available on GitHub: https://github.com/ZhongzeYan/2025_LifeScienceAlliance. The processed single-cell dataset files are available on Zenodo: https://doi.org/10.5281/zenodo.15791942.

## Supplementary Information

## Acknowledgements

This work was supported by grants from the National Key Research and Development Program of China Project (2021YFA1101402), the Informatization Plan of Chinese Academy of Sciences (CAS-WX2021SF-0301), and the National Science Foundation of China (82271428), and the Open Project Program of State Key Laboratory of Stem Cell and Reproductive Biology.

### Author Contributions

Z-Z Yan: conceptualization, data curation, formal analysis, validation, investigation, visualization, methodology, and writing—original draft, review, and editing.
P-P Liu: data curation, formal analysis, and writing—original draft, review, and editing.
H-Z Du: formal analysis and supervision.
G-L Chai: formal analysis, validation, and writing—review and editing.
Z-Q Teng: conceptualization, supervision, and project administration.
C-M Liu: conceptualization, supervision, funding acquisition, validation, project administration, and writing—review and editing.

### Conflict of Interest Statement

The authors declare that they have no conflict of interest.

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
