## [Reviewer comments · Life Science Alliance]

Life Science Alliance

Integrative transcriptomic analysis reveals oligodendrocyte lineage switching in multiple sclerosis

Zhongze Yan, Pei-Pei Liu, Hong-Zhen Du, Guoliang Chai, Zhao-Qian Teng, and Chang-Mei Liu

DOI: <https://doi.org/10.26508/lsa.202403150>

Corresponding author(s): *Chang-Mei Liu, Chinese Academy of Sciences*

Review Timeline:

Submission Date:	2024-11-28
Editorial Decision:	2025-01-13
Revision Received:	2025-04-01
Editorial Decision:	2025-04-30
Revision Received:	2025-06-06
Editorial Decision:	2025-06-27
Revision Received:	2025-07-04
Editorial Decision:	2025-07-08
Revision Received:	2025-07-15
Accepted:	2025-07-15

Scientific Editor: *Tim Fessenden*

Transaction Report:

January 13, 2025

Re: Life Science Alliance manuscript #LSA-2024-03150-T

Dr. Chang-Mei Liu
Institute of Zoology, Chinese Academy of Sciences
Orthopaedic Surgery
1 Beichen West Road, Chaoyang District
Baltimore 21205

Dear Dr. Liu,

Thank you for submitting your manuscript entitled "Integrative analysis of snRNA-seq datasets reveal molecular mechanisms of multiple sclerosis-associated oligodendrocyte lineage switch during disease progression" to Life Science Alliance. The manuscript was assessed by expert reviewers, whose comments are appended to this letter. We invite you to submit a revised manuscript addressing the Reviewer comments.

Thank you for this interesting contribution to Life Science Alliance. We are looking forward to receiving your revised manuscript.

Sincerely,

B. MANUSCRIPT ORGANIZATION AND FORMATTING:

Reviewer #1 (Comments to the Authors (Required)):

In this study, the authors used previously published single-nucleus RNA sequencing and machine learning algorithms to investigate the molecular signatures underlying oligodendrocyte pathology in MS. They documented an increase in stressed oligodendrocytes in MS, potentially due to their interactions with inflammatory microglia. They postulate that SIRPA-CD47 and CD74-MIF interactions may underlie this pathological cellular crosstalk, leading to microglia accumulation, oligodendrocyte loss and demyelination.

Major comments:

- The manuscript will benefit from an extensive proof-read. There are many spelling and grammar mistakes throughout, including in Figures, sometimes to the point where the meaning of the sentence is unclear. Significant details are also missing throughout.
- There is no decrease in OPCs evident in Figure 1 but rather an overall increase in OLs, which is largely accounted for by increase in stressed OLs. Please justify your conclusion with additional data on OPCs, or please adjust it.
- The authors propose that the increase in stressed OLs is a cause of demyelination, but it could also be a response to it. Please include this in the Discussion.
- The Stressed OL signature is poorly defined, please mention markers used for this in the text.
- Can DEG analysis be performed on Immune OLs?
- Can you compare the Stressed OLs between Control and MS NAWM to see if there are any differences?
- It is unclear how abnormal oligodendrocyte differentiation is implicated by pathways related to phagocytosis. Rather, it would more likely point to dysfunction of the mature OLs.
- Please list the markers used for the machine learning pipeline in the text in the same order as in the Figure and comprehensively list all markers used.
- I find the data presented in Figures 6 and 7 interesting but lacking validation.

Please provide staining of CD47, SIRPA, CD74 and MIF to validate their overexpression/knockdown in the cells.

In addition, showing colocalization of cells doesn't mean they are being phagocytosed. Please provide 3D analysis to show bona fide phagocytosis.

There is significant lack of detail and evidence regarding the FACs data in Figure 6. Please provide more details in the Methods and/or Supplemental on the gating strategy for the phagocytosis assay, including strategies used to select living cells, as well as antibodies used. Please also provide more details on your compensation strategies in this context.

Figure 7A is confusing and difficult to decipher due to the violet staining. Please indicate the cells of interest using arrows.

Minor comments:

- Abstract: Line 25 has a typo (in vivo instead of in vitro); Lines 27-30 are not clear at all in their meaning. Please revise extensively.
- Introduction: Reduce and refine. The neuronal information is not very relevant for this paper. Please include a section on what is already known about oligodendrocyte heterogeneity in MS.
- Fig. 1: Include proportions from Figure 1 in the text. Can statistics also be performed?
The n number for grey matter is not included in 1A, 2 clusters are not labelled in 1B, in 1E the colours for OPC and MG are too similar.
- Fig. 3A: Why are CA and CI not included?
- Lines 184-188: Where is the graph or table showing these quantifications mentioned?
- Line 185: XXX is a typo.
- Line 188: It's stated the NFOL were not present in NAWM, but in Fig. 3A it looks as if they are. Please clarify.
- Line 189: It is concluded that an inadequate supply of de novo OL underlies demyelination in MS. However, it could also be that they are dying e.g. due to inflammation processes, as suggested by the necroptosis pathway expressed. Please include this as a discussion point.
- Line 195: The pathways listed in the text do not align with those in Fig. 3E and F. Please adjust and also mention pre-OPC pathways in the text as well as MFOL.
- Line 207: Please include examples of genes in the inflammatory gene set and cite the paper from which it is derived here as well.
- Fig. 4C: Please indicate why CI is missing from the DC analysis.

- Line 212: Increase in inflammatory signature does not necessarily mean an increase in immune cell numbers. Please correct.
- Line 213: State which inflammatory marker.
- Lines 215-216: Which figure is this referring to? State.
- Line 217: Microglia are known to play a role in MS lesions, please adjust wording to "confirm" rather than "suggest".
- Fig. 5B and C: Why are lesion edges not included here? Explain significance of CD22 and APP, as they are not mentioned in the text.
- Discussion: Expand on how SIRPA-CD47 and CD74-MIF could be modulated therapeutically. What if these are protective responses? Speculate on potential functions of stressed OLs - are they a response to or cause of demyelination? Are they protective or detrimental?
- Line 300: This looks like a typo. Should it read "OPCs in remyelination"?
- Line 316: Microglia should be introduced as resident immune cells in the Introduction only.

Reviewer #2 (Comments to the Authors (Required)):

In this study Yan and Liu et al. use different published datasets of single nuclei/cell RNA samples of Multiple Sclerosis and control samples to study the dynamics of oligodendroglia on the lesion progression. The authors identify a decrease of OPC and OLs in MS and patterns like stressed OLs on the margins of the NAWM. The authors also identified specific molecules related to the interaction between Microglia and OLs in MS.

In general, the reviewer agrees in the need of re-using published datasets to generate a more comprehensive atlas or study in this case of the MS disease with a focus on oligodendroglia. However, the findings of Yan et al. are a confirmation of the previously published papers; as the decrease of OPCs MS, upregulation of immune programs in MS and the link between Microglia and OLs in MS (Jäkel et al. 2019, Schirmer et al. 2019, Yeung et al. 2019, Kukanja et al. 2024, Absinta et al. 2021).

Methodology in general appears OK and the reviewer finds interesting the use of machine learning methods to identify more specifically transitional states in development. However, the authors do not provide a detailed methodology and do not include any code use. Which specifically in the case of the machine learning method used would be fundamental.

Specific points;

1- The datasets used in the study are from different platforms. They are all 10X Genomics but from different versions. Also, I would expect batch effects, doublets and other technical issues when integrating them. The authors should provide all the QC metrics and detailed information on the alignment and processing. The Methods section does not include numbers and parameters. Including the code will also be helpful. It is not clear to me if the authors are using the processed count matrixes from the original publications or the data was processed from the raw fast files. In both cases the preprocessing, integration and normalization of the data should be included.

2- Through the study the authors use percentages to compare gain and loss of different populations in MS, (Figure 1 E and G), for instance. It is not very correct to compare directly proportions in single cell data, because this can be biased by the dissociation protocol, the area that was dissected, technical issues between samples ... Nowadays spatial transcriptomics would be the way to directly compare proportions of populations. In this case the authors should correct that and use methods like MILO, scCODA or others.

3- The machine learning method lacks information. The code and the model should be included. The model was trained with a small dataset. Would not be more specific to use a more complete atlas of the human brain as Braun et al. 2023 Nature. What is the reason to use a developmental dataset as training dataset to predict on an adult disease dataset? Not sure, if I am getting the reasoning to have a model in development to be applied in the adult brain and MS brain. Some of the populations will be missing due to adult brain features but others more related to the disease. Not sure if it is the best approach, can the authors comment on that?

4- A recent publication shows that the individual patients show high variability in MS when analyzing single nuclei transcriptomics data, MacNair et al. 2024 Neuron. This would mean that it is difficult to identify general mechanisms to act against the disease. Any comments about this? This study also includes some of the datasets used by the authors and makes a big point on the statistics to be able to integrate them properly.

5- Coming back to previous points, the lack of sample/individual level integration and normalization could also reflect biases towards some of the datasets used in the study. How were different patients/samples normalized together?

6- Do the results correlate with some more recent studies using spatial transcriptomics (Kukanja et al. 2024 Cell and Lerma-Martin et al. 2024 Nature Neuroscience).

7- In the line 185, but less in CA (XXX) is missing.

8- Through the text significant p-val is written as P val less than 0.05. This refers to the adjusted p-value, right?

Responses to the comments

We appreciate the editors and anonymous reviewers for your in-depth comments, suggestions and corrections, which have greatly improved our manuscript. We have addressed all the comments and revised the manuscript. The point-by-point responses (in **blue**) to each comment (in *italic*) are shown below, and all of the major revisions are highlighted in **red** in the revised manuscript.

COMMENTS TO AUTHOR:

Reviewer #1: In this study, the authors used previously published single-nucleus RNA sequencing and machine learning algorithms to investigate the molecular signatures underling oligodendrocyte pathology in MS. They documented an increase in stressed oligodendrocytes in MS, potentially due to their interactions with inflammatory microglia. They postulate that SIRPA-CD47 and CD74-MIF interactions may underly this pathological cellular crosstalk, leading to microglia accumulation, oligodendrocyte loss and demyelination.

1. *The manuscript will benefit from an extensive proof-read. There are many spelling and grammar mistakes throughout, including in Figures, sometimes to the point where the meaning of the sentence is unclear. Significant details are also missing throughout.*

Response: Thank you for your advice. We have carefully proofread the entire manuscript, correcting any grammatical mistakes and spelling errors within the text in our revised manuscript.

2. *There is no decrease in OPCs evident in Figure 1 but rather an overall increase in OLs, which is largely accounted for by increase in stressed OLs. Please justify your conclusion with additional data on OPCs, or please adjust it.*

Response: We acknowledge that batch effects across different samples can confound direct analyses based solely on cell proportions. Thus, we have employed cell abundance analysis, which offers a more robust assessment by accounting for potential batch-related variations. This data was added in New Figures 1E and G. Thus, we have adjusted our conclusions as “the abundance of stressed OLs and ImOLs increased in most of MS lesions” in our revised manuscript (*Lines 117-124, 135-136*).

3. *The authors propose that the increase in stressed OLs is a cause of demyelination, but it could also be a response to it. Please include this in the Discussion.*

Response: We appreciate the reviewer’s concerns. On the one hand, stress OLs may be the

result of demyelination: The inflammatory environment of MS (such as microglia contribute to demyelination through oxidative stress induced by reactive oxygen species) can first lead to demyelination, followed by OLs being damaged by environmental stress. On the other hand, stressed OLs may lead to demyelination: Our data also support the idea that stressed OLs upregulate necroptosis and efferocytosis related pathways and may be unable to maintain myelin due to impaired function.

Several studies have shown that OLs dysfunction, such as mitochondrial dysfunction or metabolic stress, can lead to myelin damage and axonal degeneration (López-Muguruza E & Matute C, 2023). MS-specific antibodies can cause OLs loss and demyelination, suggesting that OLs dysfunction directly contributes to demyelination (Liu Y et al, 2017). Thus, inflammation may be the initial trigger for demyelination, and OLs are subsequently damaged by environmental stress. Taken together, MS lesions result in the rise of stressed OLs, leading to demyelination, accompanied by prolonged inflammatory response, which, in turn, can promote further demyelination and the emergence of stressed OLs. We have added this discussion to our article (*Lines 348-357*).

4. *The Stressed OL signature is poorly defined, please mention markers used for this in the text.*

Response: Thanks for your suggestion. We have added this information in our new Supplemental Table S3B, and we have defined stressed OL as highly reflecting the stress due to MS immune reaction and injury in our revised manuscript (*Lines 131-134*).

5. *Can DEG analysis be performed on Immune OLs? Can you compare the Stressed OLs between Control and MS NAWM to see if there are any differences?*

Response: Yes, we performed DEG analysis on ImOLs, and we added this result in the new supplemental Table S7, and we found DEG associated with inflammatory mediator regulation, such as ASIC and ITPR1, suggesting that ImOLs mediated active immune response (*Lines 150-153*).

We performed DEG analysis and KEGG, GO enrichment analysis on Stressed OLs between NAWM and healthy controls. We found activated interaction stressed OLs connect in NAWM than healthy control (*Lines 144-148*).

6. *It is unclear how abnormal oligodendrocyte differentiation is implicated by pathways related to phagocytosis. Rather, it would more likely point to dysfunction of the mature OLs.*

Response: Thank you for your valuable suggestion. We have carefully reviewed our pathway analysis results regarding abnormal oligodendrocytes (Stress OLs and ImOLs). As you

pointed out, pathways related to apoptosis and phagocytosis are more likely to reflect functional impairments in abnormal OLs, rather than evidence of differentiation failure. These pathways suggest that abnormal OLs have dysfunctional roles in supporting mature OL function and regulating apoptotic processes. We have revised this conclusion in the manuscript (*Lines 150-153*).

7. *Please list the markers used for the machine learning pipeline in the text in the same order as in the Figure and comprehensively list all markers used.*

Response: We used the following high-confidence validation gene list to identify OPC and OL proto cell subtypes (Pre-OPC, Dividing OPC, Late-OPC, COP, NFOL, MFOL, MOL) for training our machine learning model. We have incorporated this detailed information into the revised manuscript to enhance clarity and reproducibility (*Lines 172-176*).

Pre-OPC: *EGFR, NES, ZFP36L1* and *GFAP*.

Dividing-OPC: *MKI67, ASPM, TOP2A* and *CDK1*.

Late-OPC: *SOX10, NKX2-2, PCDH15* and *CSPG4*.

COP: *NEU4, SOX6* and *GPR17*.

NFOL: *TCF7L2, ITPR2, TMEM2* and *GPR17*.

MFOL: *MAL, MOG, PLP1, OPALIN, SERINC5*.

MOLs: *KLK6, APOD, FAR1* and *PMP22*.

8. *I find the data presented in Figures 6 and 7 interesting but lacking validation. Please provide staining of CD47, SIRPA, CD74 and MIF to validate their overexpression/knockdown in the cells.*

Response: We appreciate this suggestion. We stained CD47 (new Supplemental Figure S6B-C) and MIF (new Supplemental Figure S7B-C) to confirm their overexpression in MO3.13 cells. We included staining for SIRPA and CD74 to confirm its knockdown in the HMC3 cell lines in our revised manuscript, now shown in new Supplemental Figures S6E-F and S7E-F. We have addressed this data in our revised manuscript (*Lines 260-264,281-285*).

9. *In addition, showing colocalization of cells doesn't mean they are being phagocytosed. Please provide 3D analysis to show bona fide phagocytosis.*

Response: Thank you for your suggestion. We provided three-dimensional reconstruction of confocal images. OPCs (MO3.13 lines) infected viruses exhibit green GFP fluorescence. Microglia (HMC3 lines) infected viruses exhibit red mCherry fluorescence. High-magnification confocal z-stack images (Zeiss LSM880) of OPCs (MO3.13 lines) phagocytosed by microglia (HMC3 lines) were converted to three-dimensional images using the surface and colocalization functions in Imaris software (Bitplane, version 9.0) to colocalize and reconstruct the surface. The colocalization of GFP and mCherry is regarded as

OPCs within microglia. Therefore, the volume of GFP in mCherry microglia within the field is used to calculate the engulfment quantification. We have added it to our revised manuscript in the new Figure 6A (*Lines 530-536*).

10. *There is significant lack of detail and evidence regarding the FACs data in Figure 6. Please provide more details in the Methods and/or Supplemental on the gating strategy for the phagocytosis assay, including strategies used to select living cells, as well as antibodies used. Please also provide more details on your compensation strategies in this context.*

Response: We sincerely appreciate the valuable comments. OPCs (MO3.13 lines) infected viruses exhibit green GFP fluorescence. Microglia (HMC3 line) infected viruses exhibit red mCherry fluorescence. Flow cytometry analysis used the cell's fluorescence, no antibody staining, and no modulation compensation. The detailed flow cytometry analysis is as follows: The cells were harvested and washed twice with PBS. Immediately before flow cytometric analysis, the cells were strained through a 100- μ M filter. Flow cytometry analyses were performed on the BD FACS Aria II (Becton Dickinson). Flow cytometry phagocytosis plots showing rates of engulfment of OPCs (labeled with GFP, FITC) when cocultured with HMC3 lines (labeled with mCherry, PE). Data were analyzed using FlowJo software (FlowJo). We have added it to our materials and methods in our revised manuscript (*Lines 544-553*).

11. *Figure 7A is confusing and difficult to decipher due to the violet staining. Please indicate the cells of interest using arrows.*

Response: We appreciate this suggestion. We used ImageJ to label each cell, as shown in the Response Figure 1. At the same time, we added arrows to indicate cells in our new manuscript in new Figure 7A.

[Figure removed by editorial staff per authors' request]

Minor comments:

12. *Abstract: Line 25 has a typo (in vivo instead of in vitro); Lines 27-30 are not clear at all in their meaning. Please revise extensively.*

Response: We sincerely thank the reviewer for identifying these issues and providing constructive feedback. We have carefully revised the abstract to improve the clarity and precision of the text in our new revised manuscript (*Lines 25,27-30*).

13. *Introduction: Reduce and refine. The neuronal information is not very relevant for this paper. Please include a section on what is already known about oligodendrocyte heterogeneity in MS.*

Response Thank you for your suggestion. We have revised the Introduction section by reducing and refining the neuronal information to focus more on oligodendrocyte heterogeneity in MS. Following the reviewer's suggestion, we have streamlined the Introduction to focus on the core aspects of MS pathology relevant to our study, specifically demyelination, inflammation, and remyelination fail. We have added a new paragraph summarizing the current understanding of oligodendrocyte heterogeneity in MS pathology (*Lines 52-58*).

14. *Fig. 1: Include proportions from Figure 1 in the text. Can statistics also be performed?*

Response: Thanks for your advice. It is not quite accurate to compare proportions in this integrated dataset. Because this can be biased by batch effect on each sample. To solve this problem, we use MiloR to perform a differential abundance test. This method can identify perturbations and maintain false discovery rate control across batch effects. We have added this data in our revised manuscript (*Lines 74-75, 117-119, 456-460*).

15. *The n number for grey matter is not included in 1A, 2 clusters are not labelled in 1B, in 1E the colours for OPC and MG are too similar.*

Response: Since grey matter was not involved in our analysis, the diagram only shows the location information of each lesion region. We have removed the grey matter label in the new Figure 1 to prevent confusion. We add label for AST2 and Vasc clusters in new Figure 1B. Both AST2 and Vasc clusters had fewer than three replicates and were therefore excluded from the following analysis (*Lines 119-120*). We revised the colours palette in Figure 1B to improve data visualization, ensuring that each category is clearly identifiable.

16. *Fig. 3A: Why are CA and CI not included?*

Response: In these two lesion regions, the machine learning-based approach identified very few proto cell types for OL and OPCs, reflecting the limited OL regeneration and loss of

remyelination capacity in MS lesions. Therefore, these regions were excluded from Figure 3A. We have added a corresponding explanation in the main text to clarify this point (*Lines 77-78*)

17. *Line 185: XXX is a typo.*

Response: Thanks for your attention. We have removed this typo in the revised manuscript.

18. *Line 188: It's stated the NFOL were not present in NAWM, but in Fig. 3A it looks as if they are. Please clarify.*

Response: We acknowledge that training the model with a small dataset may limit its ability to capture the full features of OL subtypes. To address this, we incorporated a broader human brain developmental atlas as the training dataset (Braun E et al, 2023, Herring CA et al, 2022, Su Y et al, 2022, van Bruggen D et al, 2022). We found NFOLs were abundantly detected in NAWM but were nearly absent in the cores of MS lesions (CA and CI) and detected in only limited numbers at lesion edges, where their abundance was lower than that observed in healthy adults. We have changed this data in our New Figure 2C and New Figure 3A (*Lines 165-169, 193-199*).

19. *Line 189: It is concluded that an inadequate supply of de novo OL underlies demyelination in MS. However, it could also be that they are dying e.g. due to inflammation processes, as suggested by the necroptosis pathway expressed. Please include this as a discussion point.*

Response: We appreciate this suggestion. During myelination, the OLs go through three canonical stages: OPCs, newly-formed OLs (de novo OLs), and mature myelinating OLs (Bergles DE & Richardson WD, 2016). Our study found a decreased presence of newly formed oligodendrocytes (de novo OLs) in MS lesions, suggesting that an insufficient supply of these cells may contribute to demyelination. MS lesions are characterized by a central vein where inflammation occurs and serum components leak out (Reich Daniel S et al, 2018). Infiltrating inflammatory cells are activated and interact with other immune cells and neuronal cells, resulting in oligodendrocyte death-mediated demyelination. We have added it to the discussion in our revised manuscript (*Lines 348-357*).

20. *Line 195: The pathways listed in the text do not align with those in Fig. 3E and F. Please adjust and also mention pre-OPC pathways in the text as well as MFOL.*

Response: We have updated Figure 3D. We also adjusted the text and added enrichment results and descriptions of Pre-OPC. (*Lines 211-213*)

21. *Line 207: Please include examples of genes in the inflammatory gene set and cite the paper from which it is derived here as well.*

Response: We have included the gene set in our new Supplementary Table S3A and added relevant citations in the revised manuscript (*Lines 219-221*).

22. *Fig. 4C: Please indicate why CI is missing from the DC analysis.*

Response: We did not identify DC cells in the CI sample, so the analysis results for this cell type are not shown in Figure 4.

23. *Line 212: Increase in inflammatory signature does not necessarily mean an increase in immune cell numbers. Please correct.*

Response: Thanks for your correction. We have revised it in our new manuscript (*Lines 234-235*).

24. *Line 213: State which inflammatory marker.*

Response: We have added inflammatory marker genes to our new Supplemental Table S3B (*Line 227*).

25. *Lines 215-216: Which figure is this referring to? State.*

Response: This conclusion is referring to Fig. 4D and Fig. S4C-D. We have modified it in the revised manuscript (*Lines 229-230*).

26. *Line 217: Microglia are known to play a role in MS lesions, please adjust wording to "confirm" rather than "suggest".*

Response: OK, we have revised it in the new manuscript (*Lines 230-232*).

27. *Discussion: Expand on how SIRPA-CD47 and CD74-MIF could be modulated therapeutically. What if these are protective responses? Speculate on potential functions of stressed OLs - are they a response to or cause of demyelination? Are they protective or detrimental?*

Response: The therapeutic modulation of the SIRPA-CD47 and CD74-MIF pathways presents a promising avenue for addressing demyelination in MS. The SIRPA-CD47 axis, known for its role in mediating “don't eat me signal” to phagocytes/microglia (Gardai SJ et al, 2005). In patients with MS, we observed a reduced CD47 expression in OL and OPCs. Experimental models suggest that CD47-SIRPα signaling could be targeted to enhance

the clearance of damaged cells or to protect neurons from premature phagocytosis (Han MH et al, 2012, Jiang D et al, 2022). In addition, the CD74-MIF pathway, implicated in immune cell migration, inflammatory responses and cell survival (Lue H et al, 2007), which can be modulated to either amplify its protective effects or to mitigate its contribution to chronic inflammation. It is reflect that these pathways represent protective responses aimed at maintaining cellular homeostasis under stress conditions. In our study, the downregulation of CD47 in stressed OLs and OPC could be an adaptive mechanism to promote their elimination by microglia, thereby causing demyelination. The interaction involving CD74-MIF may initially inhibit microglial migration and promote the phagocytosis of stressed OLs. In future MS therapies, the SIRPA-CD47 and CD74-MIF pathways could serve as potential therapeutic targets. Targeting the SIRPA-CD47 axis might help to prevent excessive microglial phagocytosis of oligodendrocytes, potentially reducing demyelination while preserving OL function. Additionally, manipulating the CD74-MIF pathway could be used to balance inflammatory responses, improving cellular survival and promoting the repair of damaged tissue.

We added it in our discussion in our revised manuscript (*Lines 359-367, 374-383*).

28. *Line 300: This looks like a typo. Should it read "OPCs in remyelination"?*

Response: Yes, we have fixed this in the new manuscript (*Lines 324*).

29. *Line 316: Microglia should be introduced as resident immune cells in the Introduction only.*

Response: Microglia are the resident immune cells of the brain (Mariani MM & Kielian T, 2009), aided in their tasks by fighting bacterial and viral infections, constantly removing debris such as dead cells, damaged neurons and protein clumps (Michell-Robinson MA et al, 2015). However, dysregulated microglia activation, leads to excessive neuroinflammation, is crucial in neurodegenerative diseases development. We have added this to the introduction in the new manuscript (*Lines 65-69*).

Reviewer #2: In this study Yan and Liu et al. use different published datasets of single nuclei/cell RNA samples of Multiple Sclerosis and control samples to study the dynamics of oligodendroglia on the lesion progression. The authors identify a decrease of OPC and OLs in MS and patterns like stressed OLs on the margins of the NAWM. The authors also identified specific molecules related to the interaction between Microglia and OLs in MS.

In general, the reviewer agrees in the need of re-using published datasets to generate a more comprehensive atlas or study in this case of the MS disease with a focus on oligodendroglia.

However, the findings of Yan et al. are a confirmation of the previously published papers; as the decrease of OPCs MS, upregulation of immune programs in MS and the link between Microglia and OLs in MS (Jäkel et al. 2019, Schirmer et al. 2019, Yeung et al. 2019, Kukanja et al. 2024, Absinta et al. 2021).

Methodology in general appears OK and the reviewer finds interesting the use of machine learning methods to identify more specifically transitional states in development. However, the authors do not provide a detailed methodology and do not include any code use. Which specifically in the case of the machine learning method used would be fundamental.

Response: We appreciate the reviewer's valuable comments. We provided relevant detailed information on machine learning in our revised manuscript (*Lines 471-485*). We uploaded the original code to GitHub (https://github.com/ZhongzeYan/2025_LifeScienceAlliance).

1- *The datasets used in the study are from different platforms. They are all 10X Genomics but from different versions. Also, I would expect batch effects, doublets and other technical issues when integrating them. The authors should provide all the QC metrics and detailed information on the alignment and processing. The Methods section does not include numbers and parameters. Including the code will also be helpful. It is not clear to me if the authors are using the processed count matrixes from the original publications or the data was processed from the raw fast files. In both cases the preprocessing, integration and normalization of the data should be included.*

Response: We appreciate this valuable suggestion. Before integrating datasets, we performed data quality control by following processing. For each sample, we retained cells with fewer than 5% mitochondrial genes and genes expressed at levels between 200 and 6000 for analysis. Next, we applied the Normalize Data function with a scaling factor of 10,000 to log-normalize the data. We then employed the FindVariableFeatures function to identify the top 2,000 highly variable genes across the cells. Afterwards, we utilized the FindIntegrationAnchors and IntegrationData functions to merge all data into a single Seurat object, eliminating batch effects from various samples.

We have included the comprehensive details and parameters utilized for data preprocessing in the methods section on data processing and integration. (*Lines 433-439*).

2- *Through the study the authors use percentages to compare gain and loss of different populations in MS, (Figure 1 E and G), for instance. It is not very correct to compare directly proportions in single cell data, because this can be biased by the dissociation protocol, the area that was dissected, technical issues between samples ... Nowadays spatial*

transcriptomics would be the way to directly compare proportions of populations. In this case the authors should correct that and use methods like MILO, scCODA or others.

Response: We thank the reviewer for their insightful comment regarding the use of percentages to compare the gain and loss of cell populations in our single-cell data. To address this concern, we have revised our analysis by implementing the miloR package (version 1.2.0), which is specifically designed to assess differential abundance in single-cell data while accounting for the compositional nature of such datasets (Dann E et al, 2022). Using miloR, we re-evaluated the abundance changes of different cell populations in the MS lesion regions to controls (*Lines 117-119*). This approach leverages a neighbourhood-based framework to test for differential abundance, mitigating biases associated with direct proportion comparisons. The updated results are now reflected in revised Figure 1F and G, and we have included a detailed description of miloR methodology in the Methods section of the manuscript (*Lines 456-460*). We have uploaded the complete analysis scripts, including the miloR implementation, to a publicly accessible GitHub repository (https://github.com/ZhongzeYan/2025_LifeScienceAlliance).

3- *The machine learning method lacks information. The code and the model should be included. The model was trained with a small dataset. Would not be more specific to use a more complete atlas of the human brain as Braun et al. 2023 Nature. What is the reason to use a developmental dataset as training dataset to predict on an adult disease dataset? Not sure, if I am getting the reasoning to have a model in development to be applied in the adult brain and MS brain. Some of the populations will be missing due to adult brain features but others more related to the disease. Not sure if it is the best approach, can the authors comment on that?*

Response: Thanks for your suggestion. Thank you for your valuable suggestion. In this study, we employed a previously established machine learning approach (La Manno G et al, 2016). We recognize that training the model on a small dataset could restrict its ability to fully capture the characteristics of OL subtypes. To address this, we incorporated a broader human brain developmental atlas as the training dataset (Braun E et al, 2023, Herring CA et al, 2022, Su Y et al, 2022, van Bruggen D et al, 2022). These datasets encompass various developmental stages, including fetal, infancy, childhood, and adolescence, during which oligodendrocytes are actively produced (*Lines 165-169*). In the adult brain, the expression of numerous genes involved in the early development of oligodendrocytes becomes challenging to detect, making it difficult for unsupervised clustering methods to identify rare subtypes of regenerating oligodendrocytes. Utilizing a high-quality OL prototype-based machine learning method, we sought to enhance the identification of OL subpopulations in adult and MS brains. This methodology has been utilized in earlier research, where developmental data served as a reference for detecting rare imGCs in adults and in the brains affected by Alzheimer's disease. (Zhou Y et al, 2022), and has proven to be very effective at distinguishing transcriptionally ambiguous cell subtypes in scRNA-seq datasets.

Certain oligodendrocyte (OL) subtypes might be missing in the adult brain as a result of typical development, while others could vanish due to disease-related changes. This is exactly

why we utilized this model to investigate the existence and changes of OL subpopulations in multiple sclerosis (MS). By contrasting these predicted subtypes with those in the developmental dataset, we gain insights into which OL populations are lost, preserved, or reactivated throughout disease progression.

4- *A recent publication shows that the individual patients show high variability in MS when analyzing single nuclei transcriptomics data, MacNair et al. 2024 Neuron. This would mean that it is difficult to identify general mechanisms to act against the disease. Any comments about this? This study also includes some of the datasets used by the authors and makes a big point on the statistics to be able to integrate them properly.*

Response: MS shows considerable variation among individuals, prompting the development of single-nucleus transcriptomics. This technology enables a more thorough characterization of variations between individuals as well as at the cellular level. Nonetheless, the integration of such highly variable datasets poses a considerable challenge. Currently, researchers are working to create detailed single-cell transcriptomic maps by collecting data from various individuals. While considering individual variability enhances specificity, the effective integration of this information without sacrificing vital insights continues to be a challenge. In light of these limitations, future progress in spatial transcriptomics and single-nucleus RNA sequencing, alongside analyses of genetic, epigenetic, and phenotypic heterogeneity, might offer new avenues for personalized treatment.

In our study, we did not include individual variability directly because of inconsistent sequencing details and methodology across datasets. Instead, we used data analysis techniques to reduce the impact of these individual differences. Although these differences are significant, current methods are mainly focused on uncovering shared mechanisms of MS. Looking ahead, predictive models that consider various influencing factors may help identify common mechanisms of the disease and advance precision medicine customized for individual patients.

We have discussed the issue in our revised manuscript (*Lines 388-393*).

5- *Coming back to previous points, the lack of sample/individual level integration and normalization could also reflect biases towards some of the datasets used in the study. How were different patients/samples normalized together?*

Response: Thank you for your insightful comment. To address the potential biases arising from the integration of different patient samples, we employed rigorous normalization and batch correction strategies to minimize dataset-specific effects. Specifically, we used Seurat anchor-based integration method to integrate and normalize the single-nucleus transcriptomic data across different samples. We have added the detailed information and parameters used for data preprocessing in the methods data processing and integration section (*Lines 433-439*).

6- *Do the results correlate with some more recent studies using spatial transcriptomics (Kukanja*

et al. 2024 Cell and Lerma-Martin et al. 2024 Nature Neuroscience).

Response: Thank you for your insightful question. Both studies utilized spatial transcriptomics, offering fresh insights into the pathological mechanisms of multiple sclerosis (MS). A notable characteristic of chronic MS is the localized inflammation observed within lesions, especially evident at the center and periphery of subcortical white matter lesions. Lerma-Martin's research examined spatial transcriptomic differences between chronic active (MS-CA) and chronic inactive (MS-CI) lesions, comparing lesion cores with control samples.

Our findings correlate with various elements of the spatial transcriptomic studies. Primarily, Lerma-Martin et al. (2024) pointed out significant transcriptomic alterations associated with distinct cell subtypes, notably oligodendrocytes (OLs), astrocytes (AS), and microglia (MC). This aligns with our emphasis on oligodendrocyte subtypes, particularly stressed OLs, as well as microglial subtypes and the relationships between OLs and immune-related cell populations.

Lerma-Martin et al. (2024) noted that cell stress-related genes, such as HSPB1, showed increased expression in oligodendrocytes (OLs) within the cortical area (CA) lesions of MS patients when compared to controls. This observation is consistent with our discovery of a greater number of stressed OLs in the regions affected by CA lesions. Moreover, we noted a marked increase in HSPB1 gene expression in stressed oligodendrocytes (OLs) at the peripheries of both chronic inflammation (CI) and CA lesions, suggesting heightened cellular stress in these regions. However, our findings on the cell differentiation-associated gene CDK18 differ from those of Lerma-Martin. They reported a general downregulation of CDK18 in OLs within both CA and CI lesions in MS patients, which aligns with the phenotype indicating inadequate oligodendrocyte differentiation in MS. Conversely, we noted an upregulation of CDK18 in stressed OLs located at the borders of CA and CI lesions, while no significant changes were observed in the core areas. This discrepancy may indicate active cell differentiation at the edges of lesions, or it might stem from variations in cell clustering resolution and the areas sampled in the tissue.

It's crucial to recognize that our study employed single-nucleus RNA sequencing (snRNA-seq), which does not offer spatial resolution. As a result, we are unable to accurately determine the spatial distribution of cell type-specific pathological alterations or link these changes to specific lesion and non-lesion areas. This distinction sets our study apart from spatial transcriptomic methodologies.

We have discussed the issue in our revised manuscript (*Lines 317-321, 388-393*).

7- *In the line 185, but less in CA (XXX)*

Response: Thanks. We have removed this typo in the revised manuscript.

8- *Through the text significant p-val is written as P val less than 0.05. This refers to the adjusted p-value, right?*

Response: Thank you for your inquiry. *P* val less than 0.05. This refers to the adjusted p-value. We have corrected it in our revised manuscript (*Lines 144, 145, 210, 213*).

Bergles DE, Richardson WD (2016) Oligodendrocyte development and plasticity. *Cold Spring Harbor perspectives in biology* 8: a020453.

Braun E, Danan-Gotthold M, Borm LE, Lee KW, Vinsland E, Lönnerberg P, Hu L, Li X, He X, Andrusivová Ž, et al (2023) Comprehensive cell atlas of the first-trimester developing human brain. *Science* 382: eadf1226. doi:10.1126/science.adf1226

Dann E, Henderson NC, Teichmann SA, Morgan MD, Marioni JC (2022) Differential abundance testing on single-cell data using k-nearest neighbor graphs. *Nature Biotechnology* 40: 245-253. doi:10.1038/s41587-021-01033-z

Gardai SJ, McPhillips KA, Frasch SC, Janssen WJ, Starefeldt A, Murphy-Ullrich JE, Bratton DL, Oldenborg P-A, Michalak M, Henson PM (2005) Cell-surface calreticulin initiates clearance of viable or apoptotic cells through *trans*-activation of Irf on the phagocyte. *Cell* 123: 321-334. doi:10.1016/j.cell.2005.08.032

Han MH, Lundgren DH, Jaiswal S, Chao M, Graham KL, Garris CS, Axtell RC, Ho PP, Lock CB, Woodard JI, et al (2012) Janus-like opposing roles of cd47 in autoimmune brain inflammation in humans and mice. *Journal of Experimental Medicine* 209: 1325-1334. doi:10.1084/jem.20101974

Herring CA, Simmons RK, Freytag S, Poppe D, Moffet JJD, Pflueger J, Buckberry S, Vargas-Landin DB, Clément O, Echeverría EG, et al (2022) Human prefrontal cortex gene regulatory dynamics from gestation to adulthood at single-cell resolution. *Cell* 185: 4428-4447.e4428. doi:10.1016/j.cell.2022.09.039

-
- Jiang D, Burger CA, Akhanov V, Liang JH, Mackin RD, Albrecht NE, Andrade P, Schafer DP, Samuel MA (2022) Neuronal signal-regulatory protein alpha drives microglial phagocytosis by limiting microglial interaction with cd47 in the retina. *Immunity* 55: 2318-2335.e2317. doi:10.1016/j.immuni.2022.10.018
- La Manno G, Gyllborg D, Codeluppi S, Nishimura K, Salto C, Zeisel A, Borm LE, Stott SRW, Toledo EM, Villaescusa JC, et al (2016) Molecular diversity of midbrain development in mouse, human, and stem cells. *Cell* 167: 566-+. doi:10.1016/j.cell.2016.09.027
- Liu Y, Given KS, Harlow DE, Matschulat AM, Macklin WB, Bennett JL, Owens GP (2017) Myelin-specific multiple sclerosis antibodies cause complement-dependent oligodendrocyte loss and demyelination. *Acta Neuropathol Com* 5: 25. doi:10.1186/s40478-017-0428-6
- López-Muguruza E, Matute C (2023) Alterations of oligodendrocyte and myelin energy metabolism in multiple sclerosis. *International Journal of Molecular Sciences* 24: doi:10.3390/ijms241612912
- Lue H, Thiele M, Franz J, Dahl E, Speckgens S, Leng L, Fingerle-Rowson G, Bucala R, Lüscher B, Bernhagen J (2007) Macrophage migration inhibitory factor (mif) promotes cell survival by activation of the akt pathway and role for csn5/jab1 in the control of autocrine mif activity. *Oncogene* 26: 5046-5059. doi:10.1038/sj.onc.1210318
- Mariani MM, Kielian T (2009) Microglia in infectious diseases of the central nervous system. *Journal of Neuroimmune Pharmacology* 4: 448-461. doi:10.1007/s11481-009-9170-6
- Michell-Robinson MA, Touil H, Healy LM, Owen DR, Durafourt BA, Bar-Or A, Antel JP, Moore CS (2015) Roles of microglia in brain development, tissue maintenance and repair.

Brain 138: 1138-1159. doi:10.1093/brain/awv066

Reich Daniel S, Lucchinetti Claudia F, Calabresi Peter A (2018) Multiple sclerosis. *New England Journal of Medicine* 378: 169-180. doi:10.1056/NEJMra1401483

Su Y, Zhou Y, Bennett ML, Li S, Carceles-Cordon M, Lu L, Huh S, Jimenez-Cyrus D, Kennedy BC, Kessler SK, et al (2022) A single-cell transcriptome atlas of glial diversity in the human hippocampus across the postnatal lifespan. *Cell Stem Cell* 29: 1594-1610.e1598. doi:<https://doi.org/10.1016/j.stem.2022.09.010>

van Bruggen D, Pohl F, Langseth CM, Kukanja P, Lee H, Albiach AM, Kabbe M, Meijer M, Linnarsson S, Hilscher MM, et al (2022) Developmental landscape of human forebrain at a single-cell level identifies early waves of oligodendrogenesis. *Developmental Cell* 57: 1421-1436.e1425. doi:10.1016/j.devcel.2022.04.016

Zhou Y, Su Y, Li S, Kennedy BC, Zhang DY, Bond AM, Sun Y, Jacob F, Lu L, Hu P (2022) Molecular landscapes of human hippocampal immature neurons across lifespan. *Nature* 607: 527-533.

April 30, 2025

Re: Life Science Alliance manuscript #LSA-2024-03150-TR

Dr. Chang-Mei Liu
Chinese Academy of Sciences
Orthopaedic Surgery
1 Beichen West Road, Chaoyang District
Baltimore 21205

Dear Dr. Liu,

Thank you for submitting your revised manuscript entitled "Integrative snRNA-seq analysis reveals oligodendrocyte lineage changes in MS progression" to Life Science Alliance. The manuscript has been seen by the original reviewers whose comments are appended below. While the reviewers continue to be overall positive about the work in terms of its suitability for Life Science Alliance, some important issues remain.

Namely, Reviewer 1 expressed concern that cell subpopulations were defined by few markers some of which may not be accurate to definitively categorize these cells. They also felt the observations on phagocytosis were not convincing. On this point we concur that reporting the MFI of the GFP signal inside microglia will offer important validation whereas additional 3D imaging data is not required. Reviewer 2 noted that important methodology is not included in the revised manuscript and associated code. In addition we concur that batch effects must be examined by showing UMAPs labelled by dataset.

Our general policy is that papers are considered through only one revision cycle; however, given the importance of the issues raised and our continued interest in this work, we are open to one additional short round of revision.

Please submit the final revision according to the points raised above, along with a point by point response to the remaining reviewer comments.

To upload the revised version of your manuscript, please log in to your account: <https://lsa.msubmit.net/cgi-bin/main.plex>
You will be guided to complete the submission of your revised manuscript and to fill in all necessary information.

- A letter addressing the reviewers' comments point by point.
- An editable version of the final text (.DOC or .DOCX) is needed for copyediting (no PDFs).
- High-resolution figure, supplementary figure and video files uploaded as individual files: See our detailed guidelines for preparing your production-ready images, <https://www.life-science-alliance.org/authors>
- Summary blurb (enter in submission system): A short text summarizing in a single sentence the study (max. 200 characters including spaces). This text is used in conjunction with the titles of papers, hence should be informative and complementary to the title and running title. It should describe the context and significance of the findings for a general readership; it should be written in the present tense and refer to the work in the third person. Author names should not be mentioned.

B. MANUSCRIPT ORGANIZATION AND FORMATTING:

Sincerely,

Reviewer #1 (Comments to the Authors (Required)):

Overall, the authors addressed most of my concerns.

However, the manuscript still needs to be proof-read as some statements still lack clarity. For example: "We found activated interaction stressed OLs connect in NAWM than healthy control." Please re-read and clarify.

Additionally, lines mentioned in the rebuttal document frequently did not match those in the manuscript, which was frustrating and hampered the review process. Please cross-check for consistency in future.

Some key points have still not been addressed:

1. Stressed OL markers are still not listed, only ImOLG. ImOLG is only defined using 3 markers, one of which is CD74, usually expressed by immune cells. SERPINA3, a classical immune/disease-associated oligodendrocyte marker, is not mentioned/included. This does not seem sufficient to accurately define a subpopulation. Stressed OL markers need to be explicitly stated both in the table and the text. Table S3B should also be restructured as it does not resemble a table.
2. I find the 3D data presented unconvincing. It does not look like phagocytosis is occurring, but rather that the entire cells are nearby each other. Please provide more convincing evidence of phagocytosis (e.g. video reconstruction) or remove this data.
3. Although textual details were provided on the flow cytometry data, some key details are still missing. It would be useful to see which size range the oligodendrocyte particles are to confirm these are indeed engulfed, or whether this is an artefact. Showing the data by mean fluorescence intensity of GFP signal inside the microglia population would be more convincing. Plots showing the backgating strategy should also be provided.

Reviewer #2 (Comments to the Authors (Required)):

The authors have improved the manuscript, but I still have some concerns on specific points. Below the authors answers with my comments;

Authors:

Response: We appreciate the reviewer's valuable comments. We provided relevant detailed information on machine learning in our revised manuscript (Lines 471-485). We uploaded the original code to GitHub (https://github.com/ZhongzeYan/2025_LifeScienceAlliance).

Reviewers answer: The repository only includes some of the scripts for Differential abundance and the ML method. All the methodology used for the datasets integration and downstream analysis is missing. I would like to see all the code used, especially the preprocessing with the dataset integration.

Authors:

1- Response: We appreciate this valuable suggestion. Before integrating datasets, we performed data quality control by following processing. For each sample, we retained cells with fewer than 5% mitochondrial genes and genes expressed at levels between 200 and 6000 for analysis. Next, we applied the Normalize Data function with a scaling factor of 10,000 to log-normalize the data. We then employed the FindVariableFeatures function to identify the top 2,000 highly variable genes across the cells. Afterwards, we utilized the FindIntegrationAnchors and IntegrationData functions to merge all data into a single Seurat object, eliminating batch effects from various samples. We have included the comprehensive details and parameters utilized for data preprocessing in the methods section on data processing and integration. (Lines 433-439).

Reviewers answer: Although the authors provide now information on how the integration was performed. It will be more convincing showing UMAPs labelled by dataset to show the level of integration with some kind of visualization. This will show an idea of the level of batch effect removal. The methods section does not include specific metrics, and the code is not included in the repository for the case of the integration.

Reviewers answer: I noticed that figure3 has been updated. In Figure 3 B the authors now show a different violinplot with the identified markers for the Oligodendrocyte lineage types. In Line 181 of the updated text, now the authors show the markers used to annotate the oligodendrocyte types, but when looking now Figure 3 B, I see that these markers do not correspond. For

instance, the case of OPALIN appearing as COP markers. OPALIN is a clear marker of mature OLs and not COPs. This gene is also used in the author's markers list. Why has this panel changed, Figure 3B? Is there a mistake on the labelling? Those are not markers for the types labelled. This statement is not true "All detected subtypes highly express their specific marker genes (Fig. 3B)"

Reviewer #1 (Comments to the Authors (Required)):

Overall, the authors addressed most of my concerns.

However, the manuscript still needs to be proof-read as some statements still lack clarity. For example: "We found activated interaction stressed OLs connect in NAWM than healthy control." Please re-read and clarify.

Additionally, lines mentioned in the rebuttal document frequently did not match those in the manuscript, which was frustrating and hampered the review process. Please cross-check for consistency in future.

Some key points have still not been addressed:

Response: We truly appreciate reviewer's helpful suggestion to improve the quality of our work. In the revised manuscript, we have rephrased these sentences for greater accuracy and readability (*page 8, line 154-156, 160-161*).

1. Stressed OL markers are still not listed, only ImOLG. ImOLG is only defined using 3 markers, one of which is CD74, usually expressed by immune cells. SERPINA3, a classical immune/disease-associated oligodendrocyte marker, is not mentioned/included. This does not seem sufficient to accurately define a subpopulation. Stressed OL markers need to be explicitly stated both in the table and the text. Table S3B should also be restructured as it does not resemble a table.

Response: Thank you for your valuable and constructive comments. In our study, the ImOLG population in our study is defined based on the co-expression of OL markers (such as MBP, MOG, and PLP1) and immune-related genes including APOE, CD74, and ITPR2. This clarification has now been added in the revised manuscript (*page 7, line 139-141*), and the complete list of markers used to define each OL subtype has been included in the updated Table S3B.

Regarding *SERPINA3*, we found that its expression was extremely low in our dataset, making it unsuitable as a reliable marker for defining stressed OLs in our samples. To further validate this observation, we examined several large-scale brain single-cell transcriptomic databases, including STAB (<https://mai.fudan.edu.cn/stab/>) and the Allen Brain Atlas (<https://portal.brain-map.org/atlasses-and-data/rnaseq>), where *SERPINA3* similarly exhibited low detection frequency. This may be due to current technical limitations such as sequencing depth.

In response to your helpful suggestion, we have restructured **Table S3B** to clearly present the markers used for each subtype.

2. I find the 3D data presented unconvincing. It does not look like phagocytosis is occurring, but rather that the entire cells are nearby each other. Please provide more convincing evidence of phagocytosis (e.g. video reconstruction) or remove this data.

Response: After additional reflection, we concur that the existing 3D visualization might not effectively illustrate phagocytosis as we hoped. To strengthen the integrity of our findings, we have opted to eliminate the 3D data from the manuscript.

3. Although textual details were provided on the flow cytometry data, some key details are still missing. It would be useful to see which size range the oligodendrocyte particles are to confirm these are indeed engulfed, or whether this is an artefact. Showing the data by mean fluorescence intensity of GFP signal inside the microglia population would be more convincing. Plots showing the backgating strategy should also be provided.

Response: We value your input on enhancing the clarity and strength of flow cytometry analysis. In this context, FSC indicates cell size, while SSC measures cellular complexity. We have now added the size distribution (FSC-A range) of the oligodendrocyte-derived particles detected in Response Figure 1. Following your suggestions, we reanalyzed the data to specifically calculate the mean fluorescence intensity (MFI) of GFP in the microglia population, as shown in New Figure 6E. To ensure precise gating, we adopted the following sequential strategy (refer to Response Figure 1 for detailed plots): First, we selected live cells: gating on FSC-A against SSC-A to exclude debris and dead cells; second, doublet exclusion: applying FSC-A versus FSC-H to remove cell aggregates and adhesion events; third, population-specific gating: using PE-Texas Red-A to identify mCherry⁺ microglia and FITC-A to differentiate GFP⁺ OPCs.

[Figure removed by editorial staff per authors' request]

Reviewer #2 (Comments to the Authors (Required)):

The authors have improved the manuscript, but I still have some concerns on specific points. Below the authors answers with my comments;

Authors:

Response: We appreciate the reviewer's valuable comments. We provided relevant detailed information on machine learning in our revised manuscript (Lines 471-485). We uploaded the original code to GitHub (https://github.com/ZhongzeYan/2025_LifeScienceAlliance).

1. Reviewers answer: The repository only includes some of the scripts for Differential abundance and the ML method. All the methodology used for the datasets integration and downstream analysis is missing. I would like to see all the code used, especially the preprocessing with the dataset integration.

Response: To address the reviewer's concern, we have re-executed and documented the preprocessing and dataset integration steps using Jupyter Notebook. We have uploaded the corresponding scripts, including those for data preprocessing, integration, and downstream analyses, as well as relevant output files, to our GitHub repository. This update ensures transparency and reproducibility of our analysis pipeline.

Authors:

1- Response: We appreciate this valuable suggestion. Before integrating datasets, we performed data quality control by following processing. For each sample, we retained cells with fewer than 5% mitochondrial genes and genes expressed at levels between 200 and 6000 for analysis. Next, we applied the Normalize Data function with a scaling factor of 10,000 to log-normalize the data. We then employed the FindVariableFeatures function to identify the top 2,000 highly variable genes across the cells. Afterwards, we utilized the FindIntegrationAnchors and IntegrationData functions to merge all data into a single Seurat object, eliminating batch effects from various samples. We have included the comprehensive details and parameters utilized for data preprocessing in the methods section on data processing and integration. (Lines 433-439).

2. Reviewers answer: Although the authors provide now information on how the integration was performed. It will be more convincing showing UMAPs labelled by dataset to show the level of integration with some kind of visualization. This will show an idea of the level of batch effect removal. The methods section does not include specific metrics, and the code is not included in the repository for the case of the integration.

Response: We sincerely thank the reviewer for this valuable suggestion. To address the reviewer's concern, we re-executed the data integration process using Jupyter Notebook and visualized the results with UMAP plots labeled by dataset of origin (see Response Figure 2 and Figure S1A). This visualization demonstrates effective batch correction and integration of the datasets. The integration workflow and corresponding batch effect removal can be found in our GitHub repository at the following link: https://github.com/ZhongzeYan/2025_LifeScienceAlliance/blob/main/LSA2025_01.integration.ipynb

[Figure removed by editorial staff per authors' request]

3. Reviewers answer: I noticed that figure3 has been updated. In Figure 3B the authors now show a different violin plot with the identified markers for the Oligodendrocyte lineage types. In Line 181 of the updated text, now the authors show the markers used to annotate the oligodendrocyte types, but when looking now Figure 3B, I see that these markers do not correspond. For instance, the case of OPALIN appearing as COP markers. OPALIN is a clear marker of mature OLs and not COPS. This gene is also used in the author's markers list. Why has this panel changed, Figure 3B? Is there a mistake on the labelling? Those are not markers for the types labelled. This statement is not true "All detected subtypes highly express their specific marker genes (Fig. 3B)"

Response: We sincerely thank the reviewer for their careful inspection of Figure 3B and for pointing out the inconsistency between the marker genes listed in the text and those shown in

the violin plot. This discrepancy resulted from an oversight in figure selection—we mistakenly included a version generated during the model testing phase, rather than the finalized one based on the defined subtype markers. We have now corrected this and plotted the violin diagram using the specific marker genes associated with each OPC and OL subtype. We observed that the cells predicted by our model as COPs did not exhibit strong expression of their defining marker genes (e.g., *NEU4*, *SOX6*), and similarly, NFOL cells showed limited expression of their corresponding prototype markers. This suggests that our model currently faces limitations in resolving transitional cell states, particularly from OL precursors to newly formed oligodendrocytes.

Several factors may contribute to this issue:

1. Overlapping gene expression among adjacent OL subtypes, due to the continuous nature of lineage progression.
2. Heterogeneity within predicted clusters, which may comprise mixtures of closely related cellular states.
3. Resolution limits of the current clustering and annotation algorithms, particularly in capturing intermediate states.

We have updated the figure and revised the corresponding statement in the manuscript. The original claim, “All detected subtypes highly express their specific marker genes,” has been modified to a more accurate description: “While most identified OL subtypes preferentially express their corresponding marker genes (Fig. 3B), the COP and NFOL populations predicted by the model did not exhibit strong expression of their expected early developmental markers. This highlighting the current resolution limitations in distinguishing intermediate states within the continuous OL differentiation trajectory.” (*page 10, line 208-212*).

We have also addressed this limitation in the Discussion section of the manuscript: “*While most subtypes were robustly detected, the model showed limited resolution in distinguishing transitional populations such as COPs and NFOLs, which may affect the interpretation of their abundance in MS samples, particularly within lesion cores.*” (*page 16, line 339-341*)

To enhance the resolution of subtype identification, particularly for transitional OL states, we are currently exploring more robust annotation strategies, including the integration of pseudo time analysis and the use of in vitro oligodendrocyte differentiation datasets.

June 27, 2025

RE: Life Science Alliance Manuscript #LSA-2024-03150-TRR

Dr. Chang-Mei Liu
Chinese Academy of Sciences
Orthopaedic Surgery
1 Beichen West Road, Chaoyang District
Baltimore 21205

Dear Dr. Liu,

Thank you for submitting your revised manuscript entitled "Integrative snRNA-seq analysis reveals oligodendrocyte lineage changes in MS progression". As you will see, Reviewer 2 is satisfied overall with the changes in place. We would be happy to publish your paper in Life Science Alliance pending final revisions necessary to meet our formatting guidelines. We invite you to consider the remaining suggestions from Reviewer 2, which are not required for publication.

- Please be sure that the authorship listing and order is correct.
- Please upload all figure files as individual ones, including the supplementary figure files; all figure legends should only appear in the main manuscript file
- Please add ORCID ID for the secondary corresponding author -- they should have received instructions on how to do so
- Please add the X and Bluesky handles of your host institute/organization as well as your own or/and one of the authors in our system
- Please upload a "clean" manuscript file, without the colored text.
- Please incorporate any points from the Conclusion section into the Discussion; we only allow a Discussion section
- Please be sure that all authors are mentioned in the authors' contributions section in the manuscript file.
- Please add your main, supplementary figure, and table legends to the main manuscript text after the references section.
- Please remove figures from the manuscript text and upload them separately
- Please remove the label A from Figure S3, since it has only one panel, and please do the same with its legend and call-out in the manuscript text.
- Please rename the "Data Access" section to "Data Availability".
- While the current title accurately conveys these findings, we suggest a reducing it for clarity, such as: "Integrative analysis of transcriptomics reveals molecular mechanisms of oligodendrocyte lineage switching in multiple sclerosis."
- The claim in the abstract on line 31 is confusing, it is unclear what is meant by "decreased as progression of lesion." The claim on lines 35-36 is also confusing, please verify if "retention" is meant instead of "recruitment".
- Please carefully check the entire manuscript text for correct grammar.

LSA now encourages authors to provide a 30-60 second video where the study is briefly explained. We will use these videos on social media to promote the published paper and the presenting author (for examples, see <https://docs.google.com/document/d/1-UWCfbE4pGcDdcgzcmiuJI2XMBJnxKYeqRvLLrLS0s8s/edit?usp=sharing>). Corresponding or first-authors are welcome to submit the video. Please submit only one video per manuscript. The video can be emailed to contact@life-science-alliance.org

A. FINAL FILES:

B. MANUSCRIPT ORGANIZATION AND FORMATTING:

Sincerely,

Reviewer #2 (Comments to the Authors (Required)):

Thank you for considering my comments.

Authors: 1- Response: We appreciate this valuable suggestion. Before integrating datasets, we performed data quality control by following processing. For each sample, we retained cells with fewer than 5% mitochondrial genes and genes expressed at levels between 200 and 6000 for analysis. Next, we applied the Normalize Data function with a scaling factor of 10,000 to log-normalize the data. We then employed the FindVariableFeatures function to identify the top 2,000 highly variable genes across the cells. Afterwards, we utilized the FindIntegrationAnchors and IntegrationData functions to merge all data into a single Seurat object, eliminating batch effects from various samples. We have included the comprehensive details and parameters utilized for data preprocessing in the methods section on data processing and integration. (Lines 433-439).

Reviewers answer 2:

Thank you for including the Jupyter notebooks with the code used to implement the analysis. I think the authors refer to Lines 449-457. The text combined with the code now gives an idea of the analysis. It would be very valuable to include the processed integrated R object in zenodo (or another repo for processed data). The authors have integrated several datasets which required time and computing power. I would be of great use and fundamental for researchers interested in the dataset to be able to use

the integrated data and be able to retrieve the identified celltypes.

Authors Response: We sincerely thank the reviewer for their careful inspection of Figure 3B and for pointing out the inconsistency between the marker genes listed in the text and those shown in 5 the violin plot. This discrepancy resulted from an oversight in figure selection-we mistakenly included a version generated during the model testing phase, rather than the finalized one based on the defined subtype markers. We have now corrected this and plotted the violin diagram using the specific marker genes associated with each OPC and OL subtype. We observed that the cells predicted by our model as COPs did not exhibit strong expression of their defining marker genes (e.g., NEU4, SOX6), and similarly, NFOL cells showed limited expression of their corresponding prototype markers. This suggests that our model currently faces limitations in resolving transitional cell states, particularly from OL precursors to newly formed oligodendrocytes. Several factors may contribute to this issue: 1. Overlapping gene expression among adjacent OL subtypes, due to the continuous nature of lineage progression. 2. Heterogeneity within predicted clusters, which may comprise mixtures of closely related cellular states. 3. Resolution limits of the current clustering and annotation algorithms, particularly in capturing intermediate states. We have updated the figure and revised the corresponding statement in the manuscript. The original claim, "All detected subtypes highly express their specific marker genes," has been modified to a more accurate description: "While most identified OL subtypes preferentially express their corresponding marker genes (Fig. 3B), the COP and NFOL populations predicted by the model did not exhibit strong expression of their expected early developmental markers. This highlighting the current resolution limitations in distinguishing intermediate states within the continuous OL differentiation trajectory." (page 10, line 208-212). We have also addressed this limitation in the Discussion section of the manuscript: "While most subtypes were robustly detected, the model showed limited resolution in distinguishing transitional populations such as COPs and NFOLs, which may affect the interpretation of their abundance in MS samples, particularly within lesion cores." (page 16, line 339-341) To enhance the resolution of subtype identification, particularly for transitional OL states, we are currently exploring more robust annotation strategies, including the integration of pseudo time analysis and the use of in vitro oligodendrocyte differentiation datasets.

Reviewers answer 3:

Thanks for correcting figure 3B. The reviewer agrees that capturing intermediate states can be a challenge when using single nuclei RNA-seq from adult brains. The violinplot now shows the top markers. I found it striking that there are not well-known canonical markers in the list. GFAP is a well-known Astrocyte marker which is known to increase in MS. EGFR and ZFP36L1 are also known Astrocyte markers. Could that be mixed population of OPCs and Astrocytes. How does PDGFRA, a classic OPC marker, look like? I would add in supplementary some typical markers for OLs like (MBP,PLP1) and PDGFRA for OPCs. To show the line between Astrocytes and OPCs.

July 8, 2025

RE: Life Science Alliance Manuscript #LSA-2024-03150-TRRR

Dr. Chang-Mei Liu
Chinese Academy of Sciences
Orthopaedic Surgery
1 Beichen West Road, Chaoyang District
Baltimore 21205

Dear Dr. Liu,

Thank you for submitting your revised manuscript entitled "Integrative transcriptomics data reveals molecular mechanisms of oligodendrocyte lineage switching".

There are some remaining issues that still need your attention before we can proceed with acceptance of this work.

- Please include an ORCID ID for the corresponding authors Zhao-Qian Teng.
- Please ensure the title on the manuscript file and the title entered into our system are consistent.
- Please consider the following suggestions to improve the Abstract:

Multiple sclerosis (MS) is a chronic disease of the central nervous system. The occurrence of MS is a phased process while its cause is still unclear. Here, by combining white matter single-nucleus transcriptomic datasets from MS and control samples, we found molecular crosstalk between oligodendrocytes (OLs) and immune cells involved in MS pathology. Using a machine learning approach, we identified oligodendrocyte precursor cells (OPC) and OL subtypes at various developmental stages. We highlighted their unique molecular characteristics and analyzed their distribution throughout development, adulthood, and in different regions impacted by MS. We also found an increased number of Pre-OPCs and newly formed oligodendrocytes (NFOLs) in normal appearing white matter (NAWM), which were scarcely detected in MS lesions. By cell communication analysis and in vitro co-culture, we found the interaction between SIRPA on microglia and CD47 on stressed oligodendrocytes was significantly reduced in MS lesions compared to NAWM, potentially preventing microglial phagocytosis of OLs. In contrast, CD74-MIF signaling between microglia and OLs was increased in lesions, which may lead to their retention around OLs.

- Please consider the following suggestions to improve the Alternative Abstract:

Integrative analysis of single-nucleus RNA sequencing with machine learning and cell communication analysis reveals molecular crosstalk between oligodendrocytes and immune cells in multiple sclerosis, identifying potential therapeutic targets.

- Please edit the entire manuscript text for proper English grammar and clarity. LSA permits the use of AI large language models as an aide to authors whose first language is not English. Authors remain fully accountable for the final text, and any such use must be noted in the Methods section.

Thank you for your attention to these final processing requirements. Please revise the manuscript and upload materials within 7 days.

Sincerely,

July 15, 2025

RE: Life Science Alliance Manuscript #LSA-2024-03150-TRRRR

Dr. Chang-Mei Liu
Chinese Academy of Sciences
Orthopaedic Surgery
1 Beichen West Road, Chaoyang District
Baltimore 21205

Dear Dr. Liu,

Thank you for submitting your Research Article entitled "Integrative transcriptomic analysis reveals oligodendrocyte lineage switching in multiple sclerosis". It is a pleasure to let you know that your manuscript is now accepted for publication in Life Science Alliance. Congratulations on this interesting work.

DISTRIBUTION OF MATERIALS:

Again, congratulations on a very nice paper. I hope you found the review process to be constructive and are pleased with how the manuscript was handled editorially. We look forward to future exciting submissions from your lab.

Sincerely,
